# CONCEPT-TRAK: UNDERSTANDING HOW DIFFUSION MODELS LEARN CONCEPTS THROUGH CONCEPT-LEVEL ATTRIBUTION

**Yonghyun Park**[1*]   **Chieh-Hsin Lai**[2]   **Satoshi Hayakawa**[3]   **Yuhta Takida**[2]
**Naoki Murata**[2]   **Wei-Hsiang Liao**[2]   **Woosung Choi**[2]   **Kin Wai Cheuk**[2]
**Junghyun Koo**[2]   **Yuki Mitsufuji**[2,3]

[1]University of Pennsylvania   [2]SONY AI   [3]Sony Group Corporation

## ABSTRACT

While diffusion models excel at image generation, their growing adoption raises critical concerns about copyright issues and model transparency. Existing attribution methods identify training examples influencing an entire image, but fall short in isolating contributions to specific elements, such as styles or objects, that are of primary concern to stakeholders. To address this gap, we introduce *concept-level attribution* through a novel method called *Concept-TRAK*, which extends influence functions with a key innovation: specialized training and utility loss functions designed to isolate concept-specific influences rather than overall reconstruction quality. We evaluate Concept-TRAK on novel concept attribution benchmarks using Synthetic and CelebA-HQ datasets, as well as the established AbC benchmark, showing substantial improvements over prior methods in concept-level attribution scenarios. We further demonstrate its versatility on real-world text-to-image generation with compositional and multi-concept prompts.

## 1 INTRODUCTION

Diffusion models (Lai et al., 2025; Ho et al., 2020; Song et al., 2021a;b; Rombach et al., 2022; Ramesh et al., 2022; Saharia et al., 2022) have achieved remarkable success in image generation, not merely through generating high-fidelity images, but through their ability to learn and flexibly compose concepts from training data (Rombach et al., 2022; Okawa et al., 2023).

This capability raises important questions about accountability and transparency. When models learn and exploit specific concepts from training data, stakeholders need to understand which training samples contributed to those concepts: whether for recognizing artistic contributions, ensuring fair compensation, safety auditing, model debugging, or copyright compliance (Wen et al., 2024; Somepalli et al., 2023b; Carlini et al., 2023; Somepalli et al., 2023a; GenLaw2024; Brittain, 2023).

To address these diverse needs, data attribution methods (Koh and Liang, 2017; Park et al., 2023a; Ghorbani and Zou, 2019) have emerged as promising tools, estimating how much each training example contributes to a generated output (Deng et al., 2023). These methods have proven valuable for tasks such as data valuation (Jia et al., 2019a), curation (Min et al., 2025), and understanding model behavior (Ruis et al., 2024). While recent work has begun exploring attribution methods tailored for diffusion models (Zheng et al., 2024; Lin et al., 2025; Mlodozeniec et al., 2024; Wang et al., 2024b), these approaches generally estimate contributions at the level of entire images. This broad perspective poses a critical limitation: in many practical scenarios, stakeholders care about specific concepts within an image, rather than the whole composition.

For example, consider an AI-generated image depicting an IP-protected character (e.g., Pikachu) rendered in a pencil drawing style as shown in Figure 1(a). In such cases, copyright concerns from IP holders (e.g., The Pokémon Company) would primarily focus on the character itself, not

---

*Work done during an internship at SONY AI. Email: `park19@seas.upenn.edu`

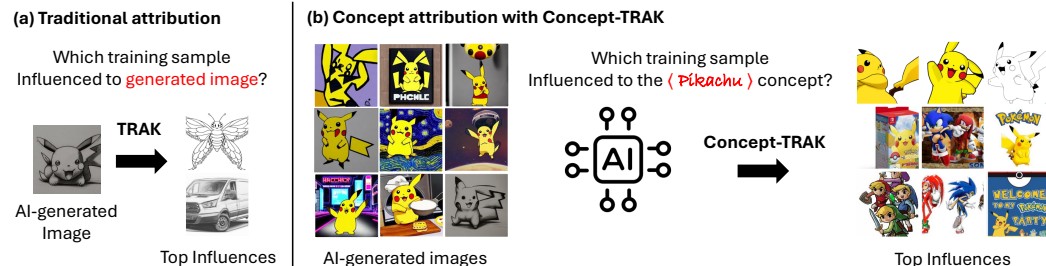

Figure 1: (a) Traditional attribution methods like TRAK identify training samples that influenced an entire generated image, often yielding influences unrelated to specific concepts of interest. (b) Our *Concept-TRAK* identifies training samples that specifically influenced a targeted concept (e.g., "Pikachu"), enabling precise attribution for features of interest.

the stylistic pencil rendering. Yet, traditional attribution methods, e.g., TRAK, identify training samples that influenced the generation of the full image, failing to isolate those tied to particular concepts. As Figure 1(a) demonstrates, these methods tend to retrieve stylistically similar images (e.g., pencil-drawn objects) but miss the character that is actually subject to copyright protection.

To address this gap, we introduce **concept-level attribution**, which estimates each training example's contribution to specific semantic features such as styles, objects, or concepts. Building on this, we propose *Concept-TRAK*, an extension of influence functions (Koh and Liang, 2017) that quantifies how training data affects the model's ability to generate *individual concepts*. Our key insight is that effective concept attribution requires designing loss functions that capture concept-relevant directions rather than optimizing for overall reconstruction quality. Concept-TRAK achieves this through reward-based formulations that explicitly target concept-relevant influences. As shown in Figure 1(b), Concept-TRAK correctly identifies training samples responsible for the concept of *Pikachu*, rather than irrelevant stylistic cues.

To rigorously evaluate our method, we introduce novel concept-level attribution benchmarks on Synthetic and CelebA-HQ datasets. Concept-TRAK substantially outperforms baselines, especially for out-of-distribution samples with unseen concept combinations. Additionally, we evaluate on the established AbC benchmark (Wang et al., 2023b), a retrieval-based framework for text-to-image model data attribution, where Concept-TRAK significantly outperforms prior methods by accurately retrieving training examples that influence specific concepts. Finally, case studies on IP-protected content, unsafe concept detection, model debugging, and relational concept learning highlight Concept-TRAK's practical utility for real-world applications and understanding concept learning in diffusion models.

## 2 BACKGROUND

### 2.1 DIFFUSION MODEL

Diffusion models (Sohl-Dickstein et al., 2015; Song et al., 2021b; Ho et al., 2020) are a class of generative models that synthesize images through an iterative denoising process. Starting from a clean image $x_0$, the forward process adds Gaussian noise to produce a sequence of increasingly noisy images $x_t$, following, $q(x_t \mid x_0) = \mathcal{N}(\sqrt{\alpha_t}x_0, (1 - \alpha_t)I)$, where $\alpha_t$ is a noise schedule controlling the level of corruption at timestep $t$.

A neural network $\epsilon_\theta(x_t, t)$ is trained to predict the added noise $\epsilon$, enabling reconstruction of $x_0$ from $x_t$ at noise level $t$. The training objective is called the denoising score matching (DSM) loss: $\mathcal{L}_{\text{DSM}}(x_0; \theta) = \mathbb{E}_{x_0, t, \epsilon}[\|\epsilon - \epsilon_\theta(x_t, t)\|_2^2]$, which encourages the model to approximate the gradient of the log-density (i.e., score function): $\epsilon_\theta(x_t, t) \propto \nabla \log p_t(x_t)$. For simplicity, we omit the timestep $t$ in $\epsilon_\theta(x_t, t)$ when the context is clear.

Diffusion models can be extended to conditional generation by incorporating additional information $c$, such as a text prompt (Ho and Salimans, 2022; Nichol et al., 2021). In this setting, the model learns $\epsilon_\theta(x_t; c) \propto \nabla \log p_t(x_t \mid c)$, allowing it to generate images that are not only realistic but also aligned with the conditioning input.

## 2.2 DATA ATTRIBUTION

The goal of data attribution is to estimate the contribution of a training sample $x^i$ to a model's utility loss $\mathcal{V}$, e.g., a performance metric or objective function that quantifies how well the model performs (e.g., test loss) (Koh and Liang, 2017; Park et al., 2023a; Ghorbani and Zou, 2019). While Leave-One-Out retraining provides exact attribution, it is computationally prohibitive for modern large-scale models.

To address this limitation, influence functions (Koh and Liang, 2017) efficiently approximate the effect of removing a training example $x^i$ using gradient-based estimates. Given a model with parameters $\theta \in \mathbb{R}^d$ and training loss $\mathcal{L}(\cdot; \theta)$, the influence function is defined as:

$$\mathcal{I}(x_0^i, \mathcal{V}) \triangleq g_{\mathcal{V}}^\top \mathbf{H}^{-1} g_i.$$

Here, $g_i = \nabla_\theta \mathcal{L}(x_0^i; \theta)$ represents the gradient of the loss with respect to parameters $\theta$ for sample $x_0^i$, $g_{\mathcal{V}} = \nabla_\theta \mathcal{V}(\theta)$ is the gradient of utility loss $\mathcal{V}$, and $\mathbf{H} = \nabla_\theta^2 \mathcal{L}(D; \theta)$ denotes the Hessian matrix of the training loss computed over the entire training dataset $D = \{x_0^i\}_{i=1}^N$. However, computing influence functions remains computationally challenging, as each attribution query requires recomputing training gradients for the entire training set in addition to the expensive Hessian computation (Choe et al., 2024).

To address this, TRAK (Park et al., 2023a) proposes projecting gradients into a lower-dimensional space using a random projection matrix $P \in \mathbb{R}^{d \times k}$ with $k \ll d$:

$$\mathcal{I}(x_0^i, \mathcal{V}) \triangleq (P^\top g_{\mathcal{V}})^\top \mathbf{H}_P^{-1} P^\top g_i,$$

where $\mathbf{H}_P = P^\top \mathbf{H} P \in \mathbb{R}^{k \times k}$ is the projected Hessian. This enables efficient storage and reuse of gradient for multiple attribution queries.

## 2.3 DATA ATTRIBUTION FOR DIFFUSION MODELS

Prior work on diffusion models (Xie et al., 2024; Zheng et al., 2024; Lin et al., 2025) has primarily focused on whole-image attribution, typically using the same objective for both training and utility losses. These studies reveal that attribution performance is highly sensitive to the choice of loss function. In particular, the standard DSM loss introduces stochasticity via both the noise term $\epsilon$ and the perturbed input $x_t$, resulting in noisy gradient estimates that require extensive averaging to be reliable, making it suboptimal for attribution. To mitigate this, D-TRAK (Zheng et al., 2024) employs the squared $\ell_2$-norm $\|\epsilon_\theta\|_2^2$, and DAS (Lin et al., 2025) employs the squared $\ell_l$-norm $\|\epsilon_\theta\|_1^1$ to compute influence scores, achieving improved stability and accuracy.

These findings highlight that **choosing a robust loss function for gradient computation is essential for reliable attribution in diffusion models**. We extend this insight to the concept-level setting, which demands loss functions specifically designed to capture concept-specific influence.

## 3 METHOD

We present *Concept-TRAK*, a framework for quantifying the contribution of individual training samples to specific concepts learned by diffusion models. Unlike prior work that measures influence on entire generated images, our approach targets how training data affects the model's ability to represent particular semantic concepts.

### 3.1 DEFINITION: CONCEPT-LEVEL ATTRIBUTION

We define concept-level attribution as measuring how training sample $x_0^i$ influences the model's ability to generate concept $c_{\text{target}}$. We quantify this through the expected concept presence:

$$p_\theta(c_{\text{target}}) = \mathbb{E}_{x_0 \sim p_{\text{sample}}(\cdot|c)} \left[ p(c_{\text{target}}|x_0) \right],$$

where $p(c_{\text{target}}|x_0)$ represents the probability that concept $c_{\text{target}}$ is present in image $x_0$, and $p_{\text{sample}}(\cdot|c)$ represents the sampling distribution used for generation.

More specifically, we consider two attribution scenarios (Figure 2):

**(a) Global concept attribution**

$\langle dog \rangle$

General concept

Q: Which training sample Influenced to the $\langle dog \rangle$ concept?

**Concept-TRAK**

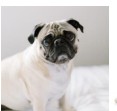 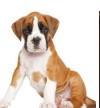 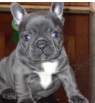 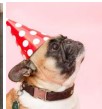

Top Influences

- - - - - - - - - - - - - - - - - - - - - - - - - - - - - - - - - - - - - - - -

**(b) Local concept attribution**

Generated image

Q: Which training sample Influenced to the $\langle dog \rangle$ concept in this image?

**Concept-TRAK**

Top Influences

Figure 2: (a) Global concept attribution identifies training samples that influenced the learning of general concepts across all generations. (b) Local concept attribution identifies training samples that influenced the learning of specific concept manifestations appearing in a particular generated image. For example, when applying local concept attribution to the "dog" concept in a generated image of a bulldog-like dog, we can observe that it retrieves images similar to bulldogs, demonstrating more targeted attribution.

- **Global attribution**: $p_{\text{sample}}$ represents the model's generative distribution (e.g., conditional sampling with CFG), measuring concept probability across all generations
- **Local attribution**: $p_{\text{sample}} = \delta(x_0 - x_0^{\text{test}})$, measuring how training data influences the specific manifestation of a concept in a particular generated image $x_0^{\text{test}}$

To approximate how training samples contribute to this concept probability, we employ influence function frameworks:

$$\mathcal{I}(x_0^i, c_{\text{target}}) = \nabla_\theta \mathcal{L}_{\text{concept}}^\top(c_{\text{target}}; \theta) \mathbf{H}^{-1} \nabla_\theta \mathcal{L}_{\text{train}}(x_0^i; \theta), \tag{1}$$

where $\mathcal{L}_{\text{concept}}(c_{\text{target}}; \theta)$ measures model performance for specific concept generation (i.e., our utility loss $\mathcal{V}$), and $\mathcal{L}_{\text{train}}(x_0^i; \theta)$ captures the training sample's contribution.

In this work, we focus on concepts $c_{\text{target}}$ that can be specified as conditioning inputs (e.g., text prompts of "Pikachu", "Mario", or class index), enabling us to leverage existing conditional generation mechanisms for precise attribution. For general concept attribution including visual concepts, please refer to the Appendix C.

## 3.2 LOSS FUNCTION DESIGN

While the theoretical setup would be to use $p_\theta(c_{\text{target}})$ as the utility loss and the standard DSM loss for training loss, prior work has shown that attribution performance is highly sensitive to loss function design (Section 2.3). The key challenge lies in designing loss functions $\mathcal{L}_{\text{concept}}$ and $\mathcal{L}_{\text{train}}$ that provide robust, concept-relevant gradients rather than generic denoising signals.

**Geometric Motivation** Our approach is motivated by the hypothesis that meaningful concept directions correspond to tangent vectors of the diffusion model's latent space, which we leverage to design concept-aware loss functions.

The latent variables of diffusion models $x_t$, lie on a lower-dimensional manifold (Chung et al., 2022b; Liu et al., 2022). Prior work has identified semantically rich structure within this manifold's tangent space, enabling concept-based editing approaches (Park et al., 2023b). Additionally, classifier-free guidance vector $\epsilon_\theta(x_t, c) - \epsilon_\theta(x_t)$, which contain rich concept information (Brack et al., 2023; Wang et al., 2023c), have been shown to operate effectively in the tangent space of the data manifold (Chung et al., 2024; Kwon et al., 2025).

Motivated by these findings, we hypothesize that concept-relevant directions can be more effectively captured by operating in the tangent space (Chung et al., 2022a; Park et al., 2023b; Wang et al., 2023c) rather than through standard denoising objectives.

**Solution: Reward Optimization**  Our geometric motivation raises a practical question: how do we identify concept-relevant directions within the tangent space and incorporate them into loss functions? We propose that reward optimization provides this capability: reward gradients $\nabla_{x_t} R(x_t)$ serve as concept-specific guidance directions that point toward concept-enhancing regions in the tangent space.

Consider the reward optimization objective (Jaques et al., 2017; Rafailov et al., 2023):

$$\max_{p_\theta} \mathbb{E}_{x_0 \sim p_\theta(\cdot|c)}[R(x_0)] - \beta \mathcal{D}_{\text{KL}}(p_\theta(\cdot|c) \| p_{\text{sample}}(\cdot|c))$$

where $R(x_0)$ is a reward function, $p_\theta(\cdot|c)$ is our target model for reward optimization, and $\beta$ controls regularization strength. While the sampling distribution $p_{\text{sample}}(\cdot|c)$ can vary in practice, we use $p_0(\cdot|c)$ in our derivation for theoretical clarity. (For general case, please refers to Appendix B.1)

The optimal solution is (Rafailov et al., 2023; Korbak et al., 2022):

$$p^*(x_0|c) \propto p_0(x_0|c) \exp(R(x_0)/\beta),$$

For diffusion models, we can extend this to intermediate timesteps by defining rewards on noisy latents $R(x_t)$ (Wallace et al., 2024).

To analyze the gradient direction toward this reward-shaped distribution, we define a loss based on Explicit Score Matching (ESM) (Vincent, 2011; Huang et al., 2021):

$$\mathcal{L}_{\text{ESM}}(x_0; \theta) = \mathbb{E}_{x_t \sim q(x_t|x_0)} \left[ \| \nabla_{x_t} \log p(x_t|c) - \nabla_{x_t} \log p^*(x_t|c) \|_2^2 \right].$$

For all subsequent loss functions, we assume $x_t \sim q(x_t|x_0)$ unless otherwise specified.

The score function of $p^*(x_t|c)$ decomposes as $\nabla_{x_t} \log p^*(x_t|c) = \nabla_{x_t} \log p_0(x_t|c) + 1/\beta \cdot \nabla_{x_t} R(x_t)$. Converting to diffusion model notation, this leads to our reward-based loss function:

$$\mathcal{L}_{\text{reward}}(x_0; \theta) = \mathbb{E}_{x_t} \left[ \| \text{sg}[\epsilon_\theta(x_t; c) - 1/\beta \cdot \nabla_{x_t} R(x_t)] - \epsilon_\theta(x_t; c) \|_2^2 \right]^{[1]}. \tag{2}$$

where $\text{sg}[\cdot]$ is stop-gradient operation, and $\beta$ is a hyperparameter whose specific choice becomes irrelevant due to gradient normalization (Section 3.4).

Intuitively, Eq. (2) steers the model's output in the direction of the reward gradient $\nabla_{x_t} R(x_t)$. We now instantiate this framework with concrete reward designs that ensure the gradients operate in the tangent space.

## 3.3 CONCEPT-TRAK

We now instantiate our reward-based framework by designing specific reward functions for concept attribution. Following Eq. (2), we replace the general reward $R(x_t)$ with two concept-specific rewards: one that increases the probability of generating training sample $x_0^i$, and another that increases the probability of concept $c$. These become our training and utility losses, respectively.

**Training Loss**  To capture how training sample $x_0^i$ influences the model's generation, we define:

$$R_{\text{train}}(x_t) \triangleq \log p(x_0^i|\hat{x}_0),$$

where $\hat{x}_0 = \mathbb{E}[x_0|x_t]$ is the posterior mean predicted by diffusion model. This reward encourages the model to generate samples likely to have originated from $x_0^i$.

Following the approach from Diffusion Posterior Sampling (DPS) (Chung et al., 2022a), we assume Gaussian distributions for the training data, i.e., $p(x_0^i|x_0) \propto \exp(-\|x_0^i - x_0\|^2/\sigma_{\text{data}}^2)$, giving us $R_{\text{train}}(x_t) \propto -1/\sigma_{\text{data}} \cdot \|x_0^i - \hat{x}_0\|^2$. While DPS uses this for posterior sampling, we propose to use it for attribution by constructing a training loss. This gradient $\nabla_{x_t} \|\hat{x}_0 - x_0^i\|^2$ operates as tangent vectors on the data manifold (Chung et al., 2022a), aligning with our geometric framework.

Substituting this into our framework (Eq. (2)) gives us the training loss:

$$\mathcal{L}_{\text{train}}(x_0; \theta) = \mathbb{E}_{x_t} \left[ \| \text{sg}[\epsilon_\theta(x_t; c) + \lambda_t \cdot \nabla_{x_t} \|\hat{x}_0 - x_0^i\|^2] - \epsilon_\theta(x_t; c) \|_2^2 \right], \tag{3}$$

---

[1] Interestingly, this tangent space motivated formulation yields an equivalent loss to $\nabla$-DB from the GFlowNet framework (Liu et al., 2025) under specific assumptions.

where $\lambda_{\text{t}} = 1/(\beta\sigma_{\text{data}})$ is a hyperparameter whose specific choice becomes irrelevant due to gradient normalization (Section 3.4).

While DSM loss has a similar goal of capturing how training samples influence generation, our train loss differs in how the learning signal is constructed. DSM provides reconstruction-driven signal, whereas our DPS-based reward explicitly yields tangent-space guidance vectors, which we empirically find more stable for concept-level attribution.

**Utility Loss**  To measure concept presence, we define:

$$R_{\text{concept}}(x_t) \triangleq \log p(c_{\text{target}}|x_t)$$

Maximizing this reward corresponds to maximizing a lower bound of our target concept probability $p_\theta(c_{\text{target}}) = \mathbb{E}_{x_0 \sim p_{\text{sample}}}[p_\theta(c_{\text{target}}|x_0)]$ (Appendix B.2). Here, $p_{\text{sample}}$ determines the attribution scope: for global attribution, it represents the model's generative distribution; for local attribution, $p_{\text{sample}} = \delta(x_0 - x_0^{\text{test}})$.

When $c_{\text{target}}$ is a conditioning input, the reward gradient reduces to classifier-free guidance vectors $\epsilon_\theta(x_t; c_{\text{target}}) - \epsilon_\theta(x_t)$ (Ho and Salimans, 2022), which operate as concept-relevant tangent vectors (Chung et al., 2024; Brack et al., 2023). For concepts embedded within condition $c$, we use concept slider guidance $\epsilon_\theta(x_t; c) - \epsilon_\theta(x_t; c_-)$ (Gandikota et al., 2024) to measure the target concept's contribution within the context, where $c_-$ is the condition that removes the target concept (e.g., $c$: "pencil drawing of Pikachu", $c_-$: "pencil drawing").

Substituting this into our framework (Eq. (2)), the corresponding utility loss is:

$$\mathcal{L}_{\text{concept}}(c_{\text{target}}; \theta) = \mathbb{E}_{x_0, x_t}\left[\|\text{sg}[\epsilon_\theta(x_t; c) + \lambda_{\text{c}} \cdot (\epsilon_\theta(x_t; c) - \epsilon_\theta(x_t; c_-))] - \epsilon_\theta(x_t; c)\|_2^2\right], \quad (4)$$

where $\lambda_{\text{c}}$ is a scaling constant whose specific value does not impact on final attribution scores due to gradient normalization (Section 3.4).

**Concept-Level Influence Function**  Having designed both training and utility losses to operate through reward gradients in the tangent space, we can now apply the influence function framework:

$$\mathcal{I}(x_0^i, c_{\text{target}}) = \nabla_\theta \mathcal{L}_{\text{concept}}(c_{\text{target}}; \theta)^\top \mathbf{H}^{-1} \nabla_\theta \mathcal{L}_{\text{train}}(x_0^i; \theta) \quad (5)$$

This measures the alignment between the guidance direction induced by training sample $x_0^i$ and the guidance direction representing target concept $c_{\text{target}}$. High alignment indicates that the training sample significantly contributed to the model's ability to generate the concept.

## 3.4 ADDITIONAL TECHNIQUES

**Deterministic Sampling via DDIM Inversion**  To eliminate stochasticity from the forward diffusion process $x_t \sim q(x_t|x_0)$ for more stable attribution, we employ deterministic DDIM inversion to derive deterministic noisy latents $x_t^i = \text{DDIMinv}(x_0^i, 0 \to t)$ from training samples $x_0^i$. Combined with our loss functions, this approach removes all sources of randomness from gradient computation, resulting in more stable influence estimates through improved gradient fidelity.

**Global vs. Local Concept Attribution**  The choice of sampling distribution $p_{\text{sample}}$ in our utility loss (Eq. (4)) determines the attribution scope. We implement this distinction as follows: (1) Global attribution uses the full conditional distribution, sampling $x_t$ via DDIM from noise. Local attribution constrains this to $p_{\text{sample}}(x_0) = \delta(x_0 - x_0^{\text{test}})$, requiring $x^{\text{test}_0} = x_0$ We enforce this constraint by sampling $x_t$ via DDIM from noise that used for generating $x_0^{\text{test}}$.

**Gradient Normalization**  Varying loss magnitudes across timesteps can cause certain gradients to dominate attribution results. To address this, we normalize each timestep gradient $g_t$ to unit norm, $\bar{g}_t = g_t/\|g_t\|_2$, ensuring that no single timestep exerts disproportionate influence on the final attribution score. This normalization also makes our method invariant to hyperparameters such as $\beta$ and $\sigma_{\text{data}}$ in our framework, providing additional robustness.

**Gradient Projection**  Following TRAK (Park et al., 2023a), we project gradients to lower-dimensional space ($k \ll d$) for computational efficiency. Moreover, we approximate the Hessian

using the Fisher Information Matrix, which requires only negligible overhead given pre-computed training gradients.

We refer to the complete method incorporating reward optimization based loss function, deterministic sampling, and gradient normalization within the influence function framework Eq. (5) as **Concept-TRAK**. Further implementation details provided in Appendix E.

## 4 EXPERIMENTS

In this section, we evaluate *Concept-TRAK* across multiple concept attribution scenarios, comparing it against TRAK (Park et al., 2023a), D-TRAK (Zheng et al., 2024), DAS (Lin et al., 2025). For text-to-image (T2I) model, we additionally compare against an unlearning-based attribution method (Wang et al., 2024b). To evaluate concept-level attribution, we conduct two controlled evaluations on class-conditional diffusion models and evaluate on an established real-world T2I model data attribution benchmark (AbC, Wang et al. (2023b)). Note that standard Linear Datamodeling Score (LDS, Park et al. (2023a)) used in traditional data attribution is inapplicable to concept-level attribution evaluation (see Appendix A for detailed discussion).

**Scope of Experiments**   Since baseline methods were not developed with concept-level attribution in mind, it is natural that they struggle under our evaluation. Our comparisons should therefore be viewed not as criticisms of prior methods, but as evidence that this newly defined task demands specialized approaches, highlighting the need for further research in this direction.

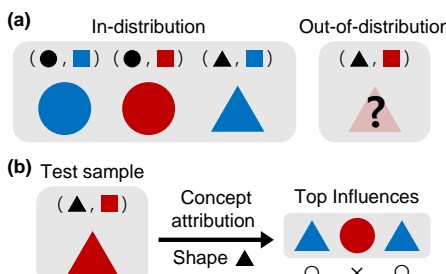

In the main text, we focused on controlled evaluations on synthetic and CelebA-HQ datasets, and evaluation on the established AbC benchmark for T2I models. In the appendix, we provide extended analyses including: (1) set-level attribution as an alternative baseline (Appendix D.1), (2) challenging real-world scenarios with semantically similar concepts (Appendix D.2.1) and complex compositional prompts (Appendix D.2.2), (3) qualitative case studies on diverse applications (Appendix D.2.3, Appendix D.4).

Figure 3: Experimental setup. (a) Train diffusion models on image–tuple pairs (shape, color), excluding all *red–triangle* combinations. (b) Generate ID/OOD samples and perform concept-level attribution; the prediction is correct if the top influential training samples contain the target concept.

### 4.1 CONTROLLED EVALUATION: SYNTHETIC DATASET

Evaluating concept attribution requires knowing the ground truth source of each concept, which is unavailable in real datasets. To address this, we construct a controlled synthetic dataset with two binary concepts: color $\in$ {red, blue} and shape $\in$ {triangle, circle}. We train a conditional diffusion model where each condition is encoded as concatenated one-hot vectors.

To comprehensively evaluate concept attribution methods, we design our dataset to enable testing in both in-distribution (ID) and out-of-distribution (OOD) scenarios (Figure 3(a)). We exclude {red, triangle} combinations from training, creating ID cases where concept combinations were seen during training (e.g., blue circle) and OOD cases requiring novel concept combinations through generalization (e.g., red triangle). This setup allows us to understand how attribution methods behave when training data directly supports the generated output versus when the model must combine concepts in novel ways.

**Evaluation Protocol**   We use Precision@10 to evaluate concept attribution. As illustrated in Figure 3(b), we generate a test image and perform concept attribution for a specific concept (e.g., "shape"). We then check whether the top-ranked training samples contain the same target concept as the generated image. In this example, a training sample with a triangle is correct ($\bigcirc$) while one with a circle is incorrect ($\times$). We generate 16 test images for each concept combination and report Precision@10 averaged across all test cases. Note that baseline methods perform standard data attribution on the generated image, while Concept-TRAK performs concept-specific attribution targeting individual concepts within that image, i.e., local concept attribution.

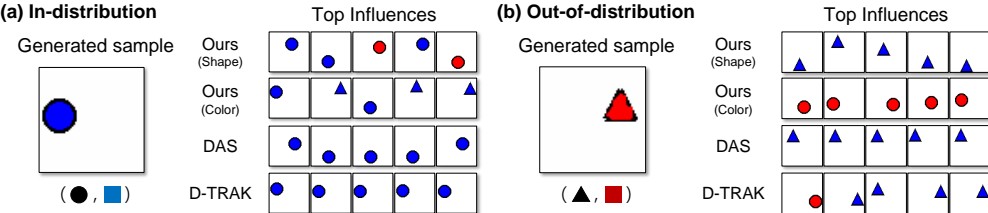

Figure 4: The target concept for each method is indicated in parentheses (Shape/Color). A data attribution method succeeds when the top influential training samples contain the same concept as the generated sample. (a) In-distribution case: Both baseline methods and our approach successfully retrieve relevant training samples. (b) Out-of-distribution: Our method accurately retrieves training samples for each individual concept (triangle for shape, red for color), while baselines can only retrieve samples related to one concept due to image-level attribution limitations.

Table 1: Precision@10 on synthetic dataset.

| | In-distribution | | | Out-of-distribution | | |
|---|---|---|---|---|---|---|
| Method | Shape | Color | Avg. | Shape | Color | Avg. |
| Ours | 1.00 | 1.00 | **1.00** | 0.80 | 0.90 | **0.85** |
| DAS | 1.00 | 1.00 | **1.00** | 1.00 | 0.00 | 0.50 |
| D-TRAK | 1.00 | 1.00 | **1.00** | 1.00 | 0.00 | 0.50 |
| TRAK | 0.67 | 0.93 | 0.80 | 0.60 | 0.30 | 0.45 |

Table 2: Precision@10 on CelebA-HQ dataset.

| | In-distribution | | | | Out-of-distribution | | | |
|---|---|---|---|---|---|---|---|---|
| Method | Eyeglasses | Male | Smiling | Avg. | Eyeglasses | Male | Smiling | Avg. |
| Ours | 0.97 | 0.93 | 0.87 | 0.92 | 1.00 | 1.00 | 0.90 | **0.97** |
| DAS | 0.99 | 0.99 | 0.90 | **0.96** | 0.70 | 0.60 | 0.70 | 0.67 |
| D-TRAK | 0.56 | 0.44 | 0.51 | 0.50 | 0.30 | 0.60 | 0.00 | 0.30 |
| TRAK | 0.86 | 0.96 | 0.71 | 0.84 | 0.60 | 0.70 | 0.50 | 0.60 |

**Results**  As shown in Table 1, Concept-TRAK maintains strong performance in both ID (1.00) and OOD (0.85) scenarios, while baseline methods exhibit a significant performance drop in the OOD setting ($\leq 0.50$). This gap underscores a fundamental distinction between attribution settings. In ID cases, image-based attribution methods can succeed for concept retrieval only by leveraging visual similarity, since there exists training samples that includes the same concept combinations as those in the generated output (Figure 4(a)). In contrast, OOD cases contain no training samples with the exact target concept combination, requiring methods to isolate the contribution of individual concepts from compositionally novel outputs (Figure 4(b)).

## 4.2 CONTROLLED EVALUATION: CELEBA-HQ

We extend our evaluation to real images using CelebA-HQ with three binary concepts: eyeglasses, male, and smiling. We deliberately exclude all samples containing the combination {eyeglasses, male, smiling} from the training dataset, creating a more challenging OOD scenario where the model should compositionally combine three concepts. We follow the same Precision@10 evaluation protocol, generating 16 test images per available combination.

**Results**  As shown in Table 2, Concept-TRAK achieves consistently strong performance in both ID (0.92) and OOD (0.97) scenarios. In contrast, DAS performs well in ID (0.96) but drops substantially in OOD settings (0.67). This difference reflects the distinct challenges of each setting: ID scenarios often benefit from image-level similarity, as retrieved samples visually resembling the generated image typically contain the target concept. However, OOD scenarios require isolating individual concepts from compositionally novel combinations, where image-level similarity is insufficient for accurate concept attribution (see qualitative results in Appendix D.3).

## 4.3 ATTRIBUTION BY CUSTOMIZATION (AbC)

We use the *Attribution by Customization* Benchmark (AbC) (Wang et al., 2023b), an established benchmark for T2I model data attribution. AbC evaluates attribution methods on models fine-tuned with exemplar images to learn new concepts via special tokens $\langle V \rangle$, measuring whether attribution method successfully retrieves the exemplars from generated images. This setup offers a rare source of ground truth: generated outputs are known to be directly influenced by the exemplars. While this setting lacks generality for large-scale training regimes, it remains the most reliable way to evaluate concept-level attribution in current T2I models.

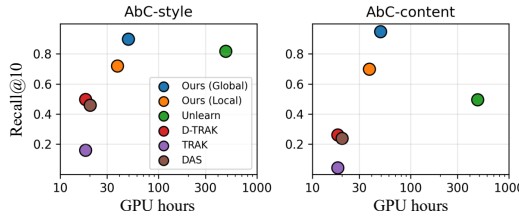

Figure 5: Recall@10 performance on AbC benchmark.

Table 3: Ablation study.

| Config | Recall@10 ($\uparrow$) |
|---|---|
| TRAK (Base: $\mathcal{L}_{\text{DSM}}$) | 0.04 |
| + (Config A: $\mathcal{L}_{\text{Reward-DPS}}$) | 0.261 |
| + (Config B: $\mathcal{L}_{\text{DPS}}$) | 0.335 |
| + (Config C: DDIMInv) | 0.564 |
| + (Config D: Normalize) | **0.955** |

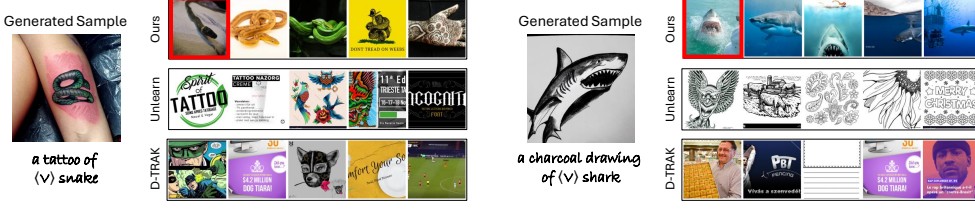

Figure 6: Qualitative results on the AbC benchmark. Correctly retrieved samples are highlighted with red boxes. Previous methods (Unlearn, D-TRAK) struggle with interference from style elements and retrieve unrelated images, while Ours (Global) successfully isolates target concepts $\langle V \rangle$.

**Evaluation Protocol**   Following the setup in Wang et al. (2023b), we report Recall@10, i.e., the proportion of times the exemplar images are successfully retrieved from a pool containing the exemplars and 100K LAION images. While the original benchmark involves not only learning exemplar using special tokens but also fine-tuning the model's parameter on the exemplar dataset, real-world use cases more commonly involve investigating the concepts generated or utilized by a single pretrained model. To better reflect this, we adopt textual inversion (Gal et al., 2022) with a frozen base model (SD1.4v) (Rombach et al., 2022), which only trains with a special token $\langle V \rangle$, without parameter updates. Further implementation details are provided in the Appendix E.

**Results**   Our method achieves significantly higher Recall@10 while maintaining computational efficiency comparable to TRAK-based method, as shown in Figure 5. As illustrated in Figure 6, prior methods often fail to isolate the concept of interest $\langle V \rangle$ due to interference from style or other visual elements in the generated image. In contrast, Concept-TRAK effectively isolates and attributes the target concept $\langle V \rangle$, demonstrating superior performance in concept-level attribution. These results can be explained by the inherently compositional nature of T2I generation. In this AbC benchmark, a model is required to combine a learned concept $\langle V \rangle$ with diverse styles or objects to generate test samples. As we demonstrated from controlled evaluation, such compositional scenario makes precise concept attribution substantially more difficult for image-based attribution methods, thereby amplifying the performance gap observed with Concept-TRAK.

**Ablation Study**   We conduct an ablation study using 48 samples from the AbC dataset to assess the impact of each design choice. Starting from the baseline TRAK with $\mathcal{L}_{\text{DSM}}$, adding concept-aware utility gradients (A), DPS-based training gradients (B), DDIM inversion (C), and gradient normalization (D) progressively improves performance, with our full method achieving 0.955 Recall@10.

## 4.4   REAL-WORLD SCENARIOS

While prior evaluations rely on controlled settings with known ground-truth concept labels, we further demonstrate the versatility of Concept-TRAK on real-world text-to-image generation using SD1.4v Rombach et al. (2022) with a 100K subset of LAION Schuhmann et al. (2022). We present representative qualitative results below; quantitative evaluation with automated concept verification and additional qualitative examples are provided in Appendix D.2.

**Compositional Prompts**   Real-world prompts often combine multiple concepts such as objects and styles. A key challenge is attributing training influence to each individual concept within a compositionally generated image. As shown in Figure 7, for a prompt *"cat in the style of graffiti art,"*

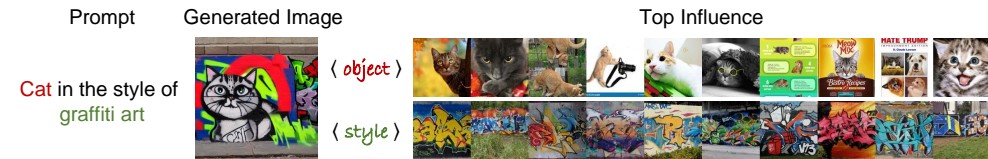

Figure 7: Qualitative results for compositional concept attribution using Concept-TRAK. For a generated image from the prompt *"cat in the style of graffiti art,"* Concept-TRAK separately retrieves training samples for the `<object>` and `<style>` concepts separately.

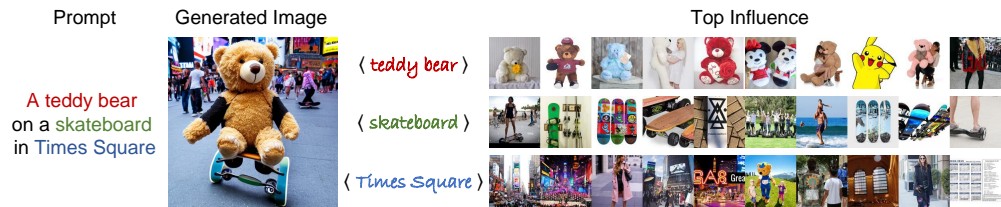

Figure 8: Qualitative results for complex multi-concept attribution using Concept-TRAK. For a generated image from the prompt *"a teddy bear on a skateboard in Times Square,"* Concept-TRAK separately retrieves training samples for each constituent concept `<teddy bear>`, `<skateboard>`, and `<Times Square>`.

Concept-TRAK successfully retrieves cat images for the object concept and graffiti art images for the style concept.

**Complex Multi-Concept Prompts**  We further evaluate on prompts that require composing three or more distinct concepts. As shown in Figure 8, for *"a teddy bear on a skateboard in Times Square,"* Concept-TRAK decomposes the prompt into its constituent concepts, separately retrieving teddy bear images, skateboard images, and Times Square images from the training data. These results demonstrate Concept-TRAK's practical utility for understanding how diffusion models compose multiple concepts from training data in real-world generation scenarios.

## 5  RELATED WORK

**Data Attribution**  Established data attribution methods include influence functions (Pruthi et al., 2020), which approximate leave-one-out retraining via gradients. TRAK and LoGra (Park et al., 2023a; Choe et al., 2024) improve scalability through random projections. Game-theoretic approaches like Data Shapley (Jia et al., 2019b; Ghorbani and Zou, 2019), based on Shapley values (Shapley et al., 1953), were initially limited by retraining costs, but recent work (Wang et al., 2023a; 2024a) improves efficiency by removing this requirement. Unlearning-based methods (Wang et al., 2024b) offer alternative trade-offs between efficiency and theoretical rigor.

**Data Attribution for Diffusion Models**  Early diffusion attribution methods adapted influence functions (Pruthi et al., 2020; Park et al., 2023a), but were biased by timestep-dependent gradient norms. Xie et al. (2024) addressed this via a re-normalized formulation. Zheng et al. (2024) extended TRAK to diffusion models, exploiting its scalability. Lin et al. (2025) later proposed the Diffusion Attribution Score, which quantifies per-sample influence by directly comparing predicted distributions, yielding more precise attributions than loss-based approaches.

## 6  CONCLUSION

In this work, we offer an initial investigation into concept-level attribution, introducing Concept-TRAK as a foundational framework. It introduces specialized reward-based training and utility loss functions designed to isolate concept-specific influences. Concept-TRAK outperforms existing methods on novel concept attribution benchmarks using the Synthetic and CelebA-HQ datasets, as well as the AbC benchmark. We expect these contributions to inspire further research toward more robust concept-level attribution benchmarks and methods for increasingly sophisticated generative models.

## STATEMENT ON THE USE OF LARGE LANGUAGE MODELS

This work made use of large language models to assist with proofreading and improving the clarity of the writing. All research ideas, theoretical development, experiments, and coding were carried out solely by the authors.

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

# A DISCUSSIONS

## A.1 LDS BENCHMARK

Linear Datamodeling Score (LDS, Park et al. (2023a)) is the most widely used benchmark in data attribution methods. LDS measures influence on specific utility losses through leave-K-out evaluation. More specifically, it linearly approximates each data point's impact on utility loss by measuring the difference between models trained on randomly subsampled data versus the full dataset.

While LDS is sometimes treated as a ground-truth proxy, it is unsuitable for our setting due to both theoretical and practical limitations:

**Structural limitation in class-conditional setting:** In datasets like CelebA-HQ, many attributes (e.g., "eyeglasses") appear in over half of the data. Under LDS's standard protocol (removing the top 50% most influential samples), many positive examples remain. Since conditional models can reproduce a concept from even a small number of positives, the measured utility drop can be minimal, systematically underestimating influence. This makes measuring each concept's true impact nearly impossible, arising from the violation of LDS's linearity assumption (Hu et al., 2024). The influence on class generation probability exhibits highly non-linear behavior, such as sharp increases after a sufficient number of sample removals.

**Computational infeasibility for text-to-image setting:** Running LDS requires retraining dozens of models (often 64+) after removing large portions of the training set. For instance, training a single T2I model on MS-COCO to reach qualitatively minimally meaningful generation already requires ∼8 GPU-days, making LDS prohibitively expensive.

**Retrieval-based evaluation as an alternative:** Our retrieval-based evaluation avoids this prevalence issue by directly checking whether the top-ranked training samples indeed contain the target concept, without being confounded by leftover positives after partial data removal.

## A.2 LIMITATIONS AND FUTURE WORK

Our research advances beyond traditional data attribution by identifying training samples that specifically contribute to particular concepts. While Concept-TRAK demonstrates superior performance compared to existing methods on concept-level attribution tasks, our approach has limitations. As illustrated in Figure 14(a) with the Simpsons example, our method occasionally retrieves stylistically similar but conceptually distinct images (e.g., other cartoon characters rather than Simpsons-specific content).

We hypothesize that this limitation stems from the fundamental challenge of gradient estimation in diffusion models. While our reward-optimization-based loss formulation and ddim inversion successfully eliminate stochasticity from the standard diffusion loss computation and provide more stable gradient estimates, perfect gradient estimation would theoretically require the true DSM loss computed with infinite Monte Carlo samples over both the noise term $\epsilon$ and noisy latents $x_t$. Our deterministic approximation, though significantly improved, cannot fully capture this infinite sampling complexity. Consequently, some attribution errors persist.

For concept-level attribution to serve as a reliable tool for addressing copyright concerns and enabling model debugging, further development is required in two key areas: (1) establishing more sophisticated benchmarks that measure concept-level attribution performance across diverse concept types and contexts, and (2) enhancing the precision and theoretical guarantees of concept-level attribution methods. This work represents an initial investigation that introduces the concept-level attribution problem and proposes Concept-TRAK as a foundational framework, and we anticipate that these contributions will catalyze further research into more robust concept-level attribution methods suitable for increasingly sophisticated generative models.

## A.3 IMPACT STATEMENTS

While our work provides tools for analyzing training data and understanding diffusion models for image generation without direct safety concerns, there exists potential for misuse by model developers who might exploit our tools to learn unsafe or problematic concepts. We emphasize that our method

is intended for responsible model development and governance, including the identification and mitigation of harmful content in training datasets, and understanding the model's behavior.

## B  THEORETICAL DETAILS

### B.1  CLASSIFIER-FREE GUIDANCE EXTENSION

We derive the reward optimization loss for the case where $p_{\text{sample}}$ corresponds to classifier-free guidance sampling. Under certain assumptions, this yields loss gradients equivalent to Eq. (2) derived from $p_0$ in the main manuscript.

Consider the reward optimization objective with classifier-free guidance sampling:

$$\max_{p_\theta} \mathbb{E}_{x_0 \sim p_\theta(\cdot|c)}[R(x_0)] - \beta \mathcal{D}_{\text{KL}}(p_\theta(\cdot|c)\|p_{\text{sample}}(\cdot|c))$$

where $p_{\text{sample}}(\cdot|c)$ represents the classifier-free guidance distribution used in practice. The optimal solution follows the same form:

$$p^*(x_0|c) \propto p_{\text{sample}}(x_0|c) \exp(R(x_0)/\beta)$$

Following the same ESM derivation as the main text, we obtain the CFG-based reward loss:

$$\mathcal{L}_{\text{reward}}^{\text{CFG}}(x_0; \theta) = \mathbb{E}_{x_t}\left[\|\text{sg}[\epsilon_\theta^{CFG}(x_t; c) - 1/\beta \cdot \nabla_{x_t} R(x_t)] - \epsilon_\theta^{CFG}(x_t; c)\|_2^2\right]$$

where $\epsilon_\theta^{CFG}(x_t; c) = \epsilon_\theta(x_t) + \gamma(\epsilon_\theta(x_t; c) - \epsilon_\theta(x_t))$ is the CFG noise prediction.

Expanding this expression:

$$\mathcal{L}_{\text{reward}}^{\text{CFG}}(x_0; \theta) = \mathbb{E}_{x_t}\left[\|\text{sg}[\epsilon_\theta(x_t) + \gamma(\epsilon_\theta(x_t; c) - \epsilon_\theta(x_t)) - 1/\beta \cdot \nabla_{x_t} R(x_t)]\right.$$
$$\left. - (\epsilon_\theta(x_t) + \gamma(\epsilon_\theta(x_t; c) - \epsilon_\theta(x_t)))\|_2^2\right]$$

The gradient of this loss is:

$$\nabla_\theta \mathcal{L}_{\text{reward}}^{\text{CFG}}(x_0; \theta) = \mathbb{E}_{x_t}\left[\frac{1}{\beta} \nabla_{x_t} R(x_t) \nabla_\theta(\epsilon_\theta(x_t) + \gamma(\epsilon_\theta(x_t; c) - \epsilon_\theta(x_t)))\right]$$

Under the assumption that $\nabla_\theta \epsilon_\theta(x_t)$ contributes minimally to concept-specific attribution directions, this simplifies to:

$$\nabla_\theta \mathcal{L}_{\text{reward}}^{\text{CFG}}(x_0; \theta) \approx \mathbb{E}_{x_t}\left[\frac{\gamma}{\beta} \nabla_{x_t} R(x_t) \nabla_\theta \epsilon_\theta(x_t; c)\right]$$

This is equivalent to the gradient of Eq. (2) up to a constant factor, which becomes irrelevant under gradient normalization. While this assumption is strong, it may be justified since $\epsilon_\theta(x_t)$ captures general denoising patterns across the dataset, which could be largely independent of specific concept directions. The effectiveness of Eq. (2) in our experiments provides some empirical support for this approximation.

### B.2  LOWER BOUND JUSTIFICATION FOR UTILITY LOSS

We provide the mathematical justification for why maximizing our reward function $R_{\text{concept}}(x_t) = \log p(c_{\text{target}}|x_t)$ corresponds to optimizing a lower bound of our target concept probability $p_\theta(c_{\text{target}}) = \mathbb{E}_{x_0 \sim p_{\text{sample}}}[p_\theta(c_{\text{target}}|x_0)]$.

Since $p_\theta(c_{\text{target}}|x_0)$ must be computed through the diffusion process, we have:

$$\log p_\theta(c_{\text{target}}) = \log \mathbb{E}_{x_0 \sim p_{\text{sample}}}[p_\theta(c_{\text{target}}|x_0)] \tag{6}$$

$$= \log \mathbb{E}_{x_0 \sim p_{\text{sample}}}\left[\mathbb{E}_{x_t \sim q(x_t|x_0)}[p_\theta(c_{\text{target}}|x_t)]\right] \tag{7}$$

$$\geq \mathbb{E}_{x_0 \sim p_{\text{sample}}, x_t \sim q(x_t|x_0)}[\log p_\theta(c_{\text{target}}|x_t)] \tag{8}$$

where the inequality follows from applying Jensen's inequality twice (due to the concavity of log). Therefore, maximizing $\mathbb{E}_{x_0, x_t}[\log p_\theta(c_{\text{target}}|x_t)]$ optimizes a lower bound of our target concept probability.

## C  OTHER TYPES OF REWARD

In this section, we show how to apply *Concept-TRAK* beyond textual concepts. We cover two scenarios: explicit differentiable reward models and implicit reward models defined by preference datasets.

### C.1  EXTERNAL DIFFERENTIABLE REWARD MODELS

Suppose we have access to an explicit, differentiable classifier that can predict the probability of a specific concept: $\log p(c_{\text{target}}|x_0)$. Here, $c_{\text{target}}$ can be any concept of interest—for example, visual features, image aesthetics, etc. If this concept classifier is trained only on clean images $x_0$, then similar to our earlier approach, we define the reward model as $R(x_t) = \log p(c_{\text{target}}|\hat{x}_0)$, where $\hat{x}_0 = \mathbb{E}[x_0|x_t]$ is the posterior mean predicted by the diffusion model (Chung et al., 2022a).

This yields our utility loss based on external reward model:

$$\mathcal{L}_{\text{Reward-DPS}}(x_t; \theta) = \|\text{sg}\left[\epsilon_\theta(x_t) - 1/\beta \cdot \nabla_{x_t} \log p(c_{\text{target}}|\hat{x}_0)\right] - \epsilon_\theta(x_t)\|_2^2.$$

### C.2  PREFERENCE DATASETS

For scenarios where concepts are defined through preference data rather than explicit classifiers, we can adapt our framework to work with preference pairs. This is particularly useful when the desired concept is subjective (e.g., aesthetic quality, safety) or difficult to define through explicit labels.

Given preference pairs $(x_0^+, x_0^-)$ where $x_0^+ \succ x_0^-$ indicates that $x_0^+$ is preferred over $x_0^-$, we need to define how a noisy latent $x_t$ sampled from our diffusion process relates to these preferences. Following the intuition that we want to steer the denoising process toward preferred outcomes and away from non-preferred ones, we define the reward function as:

$$R(x_t; x_0^+, x_0^-) = \log p(x_0^+|\hat{x}_0) - \log p(x_0^-|\hat{x}_0), \tag{9}$$

where $\hat{x}_0 = E[x_0|x_t]$ is the posterior mean predicted from the current noisy latent $x_t$.

This formulation captures the likelihood that the current denoising trajectory will lead to the preferred sample $x_0^+$ versus the non-preferred sample $x_0^-$. Under our Gaussian assumption that $p(x^i|\hat{x}_0) \propto \exp(-\|\hat{x}_0 - x^i\|_2^2)$, we obtain:

$$R(x_t; x_0^+, x_0^-) \propto -\|\hat{x}_0 - x_0^+\|_2^2 + \|\hat{x}_0 - x_0^-\|_2^2 + \text{const} \tag{10}$$

Taking the gradient with respect to $x_t$:

$$\nabla_{x_t} R(x_t; x_0^+, x_0^-) = \nabla_{x_t}\|\hat{x}_0 - x_0^-\|_2^2 - \nabla_{x_t}\|\hat{x}_0 - x_0^+\|_2^2. \tag{11}$$

This gradient naturally encourages the denoising process to move toward preferred samples $x_0^+$ (negative gradient term) and away from non-preferred samples $x_0^-$ (positive gradient term), making it suitable for integration into our Reward-DPS framework.

The resulting utility loss becomes:

$$\mathcal{L}_{\text{Preference-DPS}}((x_0^+, x_0^-); \theta) = \mathbb{E}_{x_0, x_t}[\|\text{sg}[\epsilon_\theta(x_t) - 1/\beta \cdot \nabla_{x_t} R(x_t; x_0^+, x_0^-)] - \epsilon_\theta(x_t)\|_2^2]$$

$$= \mathbb{E}_{x_0, x_t}[\|\text{sg}[\epsilon_\theta(x_t) + 1/\beta \cdot (\nabla_{x_t}\|\hat{x}_0 - x_0^-\|_2^2 - \nabla_{x_t}\|\hat{x}_0 - x_0^+\|_2^2)] - \epsilon_\theta(x_t)\|_2^2],$$

enabling concept attribution with preference data without explicitly training a reward model.

## D  ADDITIONAL RESULTS

### D.1  ADDITIONAL BASELINE: SET-LEVEL ATTRIBUTION

In the main paper, we focus our evaluation on *local concept attribution*, which measures the influence of training samples on a specific concept within a particular generated image. However, an alternative setting is *global concept attribution*, which measures the influence of training samples on the model's learned distribution over a concept in general. Unlike local concept attribution, global concept attribution can be naturally addressed by baseline methods such as DAS and D-TRAK through set-level attribution, where utility gradients are computed by averaging across multiple generated images.

**Set-level Attribution**    For a target concept $c$, we can define a set-level utility gradient as:

$$L_{\text{set}} = \mathbb{E}_{x_0 \sim p(x|c)}[L(x_0; c)] \tag{12}$$

This formulation estimates the expected loss over the model's generative distribution conditioned on concept $c$. Assuming $L$ approximates $p(x|c)$, this objective closely aligns with our concept attribution utility definition in Section 3.1:

$$p_\theta(c) = \mathbb{E}_{x_0 \sim p(x|c)}[p(c|x)] \tag{13}$$

By averaging gradients over multiple samples from $p(x|c)$, we obtain an estimate of how training data influenced the model's learned representation of concept $c$ overall, rather than its manifestation in a single image.

**Experimental Setup**    We evaluate set-level attribution across all three benchmarks (Toy, CelebA-HQ, AbC) by adapting both Concept-TRAK and baseline methods to the global attribution setting:

- **Toy & CelebA-HQ**: For each target concept $c$, we fix that concept and randomize all other attributes. We generate 256 images from $p(x|c)$ using the trained diffusion model. For each method, we compute the utility gradient as the average of individual gradients across all 256 samples.
- **AbC**: We use the benchmark-provided prompts (containing the special token) to generate 256 images. We then compute the average utility gradient across these samples.

We then rank training samples by their influence scores and evaluate whether the top-ranked samples contain the target concept using Precision@10.

**Results**    Tables 4, 5, and 6 present the results for global concept attribution across all benchmarks.

Table 4: Set-level attribution results on Toy dataset. All methods achieve strong performance on this controlled dataset.

| Concept | Concept-TRAK | DAS | D-TRAK |
|---|---|---|---|
| Shape | 1.00 | 1.00 | 1.00 |
| Color | 0.90 | 0.70 | 1.00 |
| Average | 0.95 | 0.85 | **1.00** |

Table 5: Set-level attribution results on CelebA-HQ. All methods achieve perfect performance when attributing concepts across multiple generated images.

| Concept | Concept-TRAK | DAS | D-TRAK |
|---|---|---|---|
| Eyeglasses | 1.00 | 1.00 | 1.00 |
| Male | 1.00 | 1.00 | 1.00 |
| Smile | 1.00 | 1.00 | 1.00 |
| Average | **1.00** | **1.00** | **1.00** |

The results demonstrate that Concept-TRAK achieves comparable or superior performance across all benchmarks, confirming that our design choices transfer well to the global setting. Notably, while baseline methods can perform reasonably in the global setting by averaging over many samples, they fundamentally lack the capability to perform local concept attribution for individual images—a critical limitation that Concept-TRAK addresses.

## D.2    REAL-WORLD SCENARIOS

While our controlled benchmarks (Toy, CelebA-HQ, AbC) provide rigorous evaluation with ground-truth labels, real-world applications of concept attribution often involve more challenging scenarios,

Table 6: Set-level attribution results on AbC benchmark. Concept-TRAK achieves the best average performance, with particular strength in style attribution.

| Concept | Concept-TRAK | DAS | D-TRAK |
|---------|-------------|------|--------|
| Object | 0.89 | **0.98** | 0.95 |
| Style | **0.93** | 0.81 | 0.83 |
| Average | **0.91** | 0.895 | 0.89 |

including semantically similar concepts and complex compositional prompts. In this section, we evaluate Concept-TRAK's performance on such scenarios using a large-scale text-to-image model (SD1.4v, Rombach et al. (2022)). Following AbC (Wang et al., 2023b), we use 100k subset of the LAION dataset (Schuhmann et al., 2022) for data attribution.

**Evaluation Protocol** For real-world scenarios, establishing ground-truth attribution labels is infeasible, as we cannot definitively know which training samples influenced specific concepts in generated images. However, we can perform a sanity-check evaluation by verifying whether retrieved training samples actually contain the target concept. If a method retrieves training samples that do not contain the concept being attributed, this indicates a clear failure of concept attribution.

We employ Qwen3-VL-8B (Yang et al., 2025), an open-source state-of-the-art vision-language model, to automatically assess whether retrieved images contain the target concept. For each retrieved training sample, we query the model: "Does this image contain {concept value}? Please answer with only 'yes' or 'no'" If the model responds "no", we count it as an inaccurate attribution. We report Precision@10, measuring the fraction of top-10 retrieved samples that contain the target concept.

For certain generated images, the actual number of training samples that contributed to a specific concept may be fewer than 10. Therefore, the upper bound for precision is not necessarily 1.0. These metrics should be interpreted as relative performance indicators comparing methods rather than absolute measures of attribution quality.

### D.2.1 SIMILAR CONCEPTS

A critical challenge in concept attribution is distinguishing between semantically similar concepts that share visual features. We evaluate whether methods can correctly attribute training samples for one concept without incorrectly retrieving samples from visually similar but distinct concepts.

**Experimental setup** We select five semantically similar big cat species: cat, tiger, jaguar, leopard, and cheetah. For local attribution, we generate 8 images using prompts containing only the target concept (e.g., "a photo of a cat") with different random seeds. For each generated image, we perform concept attribution and measure whether the top-10 retrieved samples contain the target concept rather than similar concepts. For global attribution, we generate 256 images per concept and compute set-level attribution by averaging utility gradients across all samples.

**Results** Tables 7 and 8 present the results for local and global attribution on similar concepts.

Table 7: Local attribution on semantically similar concepts (averaged across 8 seeds per concept).

| Concept | Concept-TRAK | DAS | D-TRAK | TRAK |
|---------|-------------|------|--------|------|
| cat | **0.925** | 0.000 | 0.000 | 0.000 |
| tiger | **0.662** | 0.000 | 0.000 | 0.000 |
| jaguar | **0.188** | 0.000 | 0.000 | 0.000 |
| leopard | **0.300** | 0.000 | 0.000 | 0.087 |
| cheetah | **0.325** | 0.000 | 0.000 | 0.000 |
| Average | **0.480** | 0.000 | 0.000 | 0.017 |

Table 8: Global attribution on semantically similar concepts (averaged over 256 generated images per concept).

| Concept | Concept-TRAK | DAS | D-TRAK | TRAK |
|---|---|---|---|---|
| cat | **1.000** | **1.000** | **1.000** | 0.900 |
| tiger | **1.000** | 0.800 | 0.800 | 0.800 |
| jaguar | **0.400** | 0.300 | 0.200 | 0.200 |
| leopard | **0.600** | **0.600** | **0.600** | 0.300 |
| cheetah | **0.500** | 0.300 | 0.300 | 0.300 |
| Average | **0.700** | 0.600 | 0.580 | 0.500 |

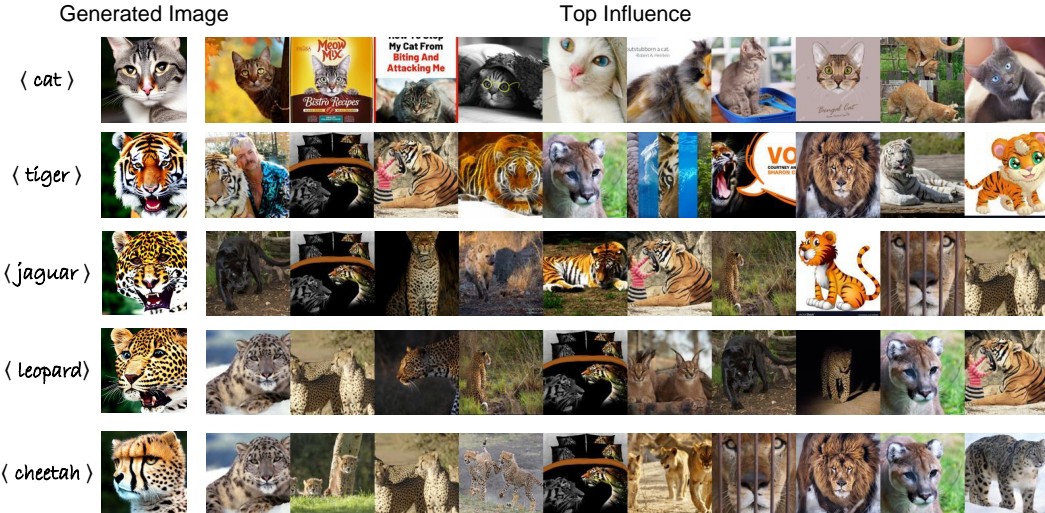

Figure 9: Qualitative results for semantically similar concepts using Concept-TRAK.

For local attribution, baseline methods achieve near-zero precision, indicating complete failure. In contrast, Concept-TRAK achieves 0.480 average precision, successfully retrieving concept-specific training samples even for visually similar classes. The performance gap is particularly pronounced for common concepts (cat: 0.925, tiger: 0.662) compared to rarer concepts (jaguar: 0.188). As shown in Figure 9, for rare concepts like jaguar, Concept-TRAK sometimes retrieves training samples of visually similar animals such as tigers with similar spotted patterns. This suggests that the model may have learned the rare concept partially through transfer from similar visual features.

For global attribution, baseline methods show improved performance by averaging over many samples, but Concept-TRAK still achieves the best results. This demonstrates that our method's design choices benefit both local and global attribution settings.

### D.2.2 COMPOSITIONAL CONCEPTS

Real-world text-to-image generation often involves compositional prompts that combine multiple concepts such as objects and styles. A key challenge is attributing training influence to each individual concept within a compositionally generated image. We evaluate this capability at two difficulty levels.

**Common Object and Style** We generate images combining common objects with artistic styles using prompts of the form "{object} in the style of {style}". We use two objects (cat, dog) and two styles (graffiti art, stained glass), generating 8 images per object-style combination for a total of 32 images. For each image, we perform separate concept attribution for the object and the style. For global attribution baselines, we generate 256 images from each full prompt and perform set-level attribution, then separately evaluate whether retrieved samples contain the object or style.

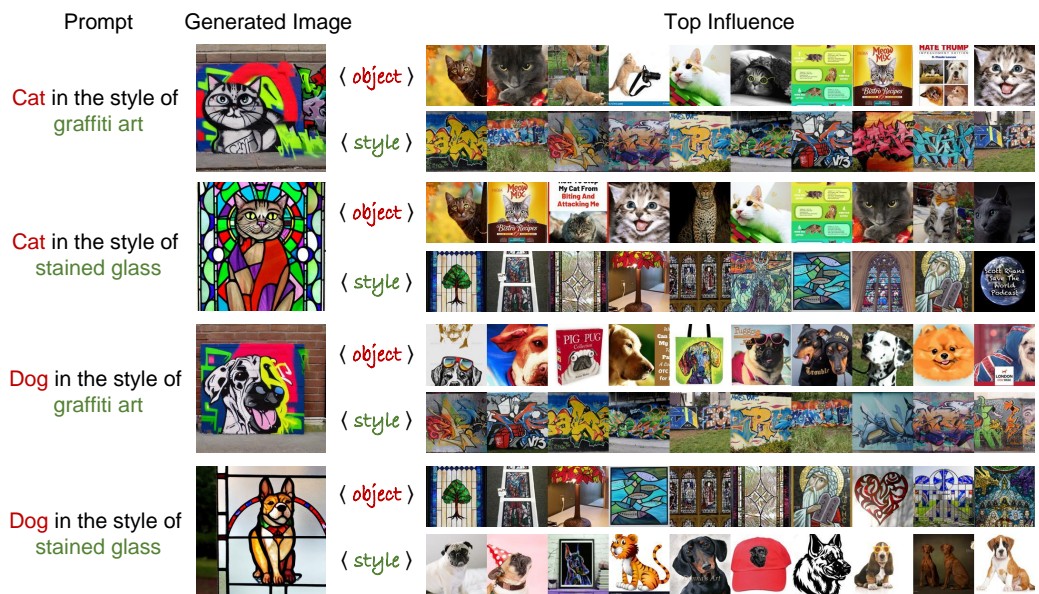

Figure 10: Qualitative results for compositional concepts using Concept-TRAK.

As Table 9 shows, Concept-TRAK substantially outperforms baselines. For local attribution, baseline methods completely fail, while Concept-TRAK achieves over 90% accuracy. For global attribution, Concept-TRAK also performs better overall. Interestingly, baseline methods appear biased toward style-based attribution, performing relatively well on global style attribution but completely failing on object attribution. For qualitative results, please refer to Figure 10.

Table 9: Compositional attribution results for natural objects with artistic styles.

| Attribution Type | Concept-TRAK | DAS | D-TRAK | TRAK |
|---|---|---|---|---|
| Local (Object) | **0.934** | 0.025 | 0.025 | 0.034 |
| Local (Style) | **0.919** | 0.047 | 0.047 | 0.013 |
| Global (Object) | **1.000** | 0.000 | 0.000 | 0.125 |
| Global (Style) | **0.950** | 0.925 | 0.925 | 0.850 |

**Unique Objects and Artist Styles.** To increase difficulty, we use unique objects (Pikachu, Simpson) and famous artist styles (Vincent van Gogh, Pablo Picasso) that require specific training data. We generate 8 seeds per combination for a total of 32 images.

Table 10: Compositional attribution results for unique objects with artist styles.

| Attribution Type | Concept-TRAK | DAS | D-TRAK | TRAK |
|---|---|---|---|---|
| Local (Object) | **0.581** | 0.000 | 0.000 | 0.000 |
| Local (Style) | **0.581** | 0.000 | 0.000 | 0.000 |
| Global (Object) | **0.725** | 0.100 | 0.125 | 0.450 |
| Global (Style) | **0.775** | 0.475 | 0.475 | 0.075 |

As shown in Table 10, Concept-TRAK successfully attributes each concept independently compared to the baseline method, even when multiple concepts are intertwined in a single image. Same as the previous experiment, baseline methods show style bias in global attribution. This suggests that using traditional data attribution methods conflates the overall visual aesthetic with style, while missing object-specific contributions. Concept-TRAK achieves balanced performance on both concepts. For qualitative results, please refer to Figure 11.

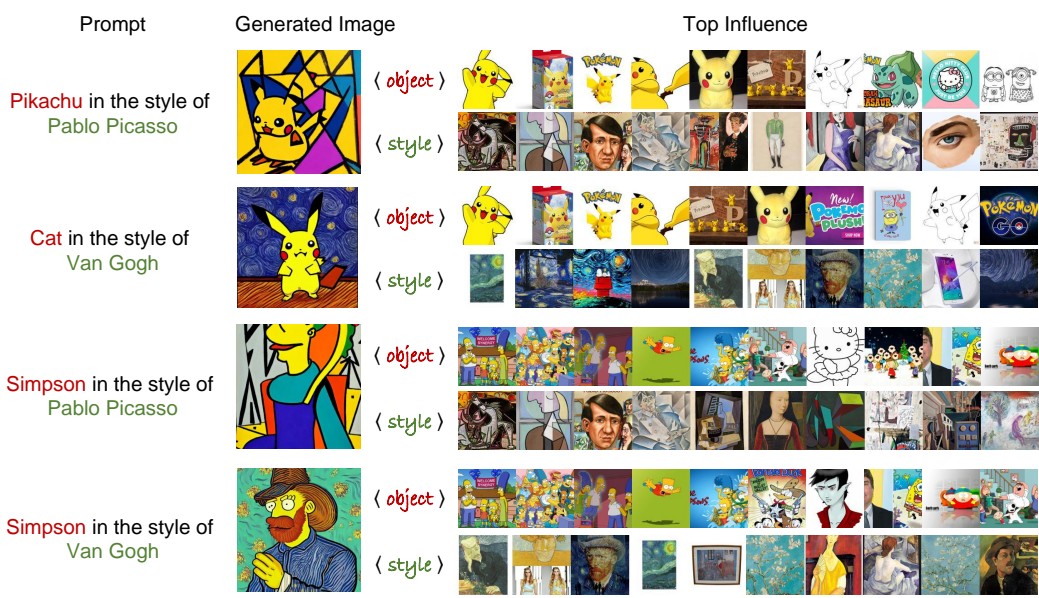

Figure 11: Qualitative results for compositional concept attribution using Concept-TRAK.

### D.2.3 COMPLEX CONCEPTS (QUALITATIVE)

Beyond quantitative evaluation, we showcase Concept-TRAK's capability on complex, multi-concept prompts commonly used in text-to-image model benchmarks. We select three challenging prompts: "An astronaut riding a horse on Mars", "A teddy bear on a skateboard in Times Square", and "Avocado chair".

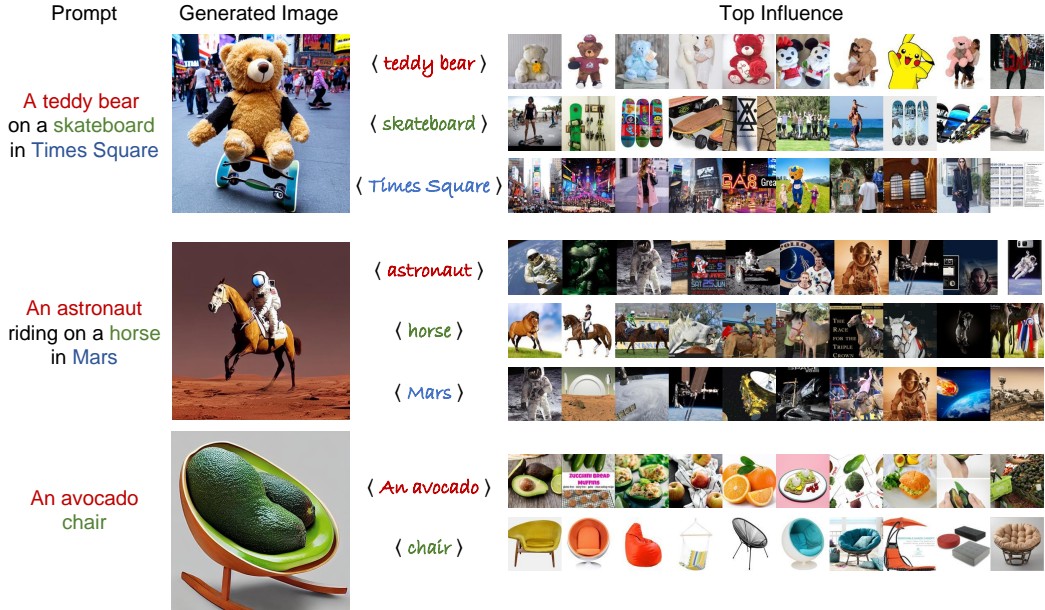

Figure 12: Qualitative results for complex multi-concept prompts using Concept-TRAK.

Figure 12 shows qualitative results for these prompts. For "astronaut riding a horse on Mars," Concept-TRAK successfully retrieves astronaut images, horse images, and Mars landscape images separately, demonstrating its ability to decompose spatially composed scenes. For "avocado chair," the method retrieves not just any chair images, but specifically round-shaped chairs that visually

Figure 13: The target concept for each method is indicated in parentheses (Eyeglasses↔Bare eyes/Female↔Male). A data attribution method succeeds when the top influential training samples contain the same concept as the generated sample. (a) In-distribution case: Both baseline methods and our approach successfully retrieve relevant training samples. (b) Out-of-distribution: Our method accurately retrieves training samples for each individual concept (eyeglasses and male), while baselines can only retrieve samples related to one concept due to image-level attribution limitations.

match the avocado-like form in the generated image. This demonstrates our local concept attribution capturing fine-grained visual features. For novel concept combinations like "avocado chair,"

These qualitative results demonstrate Concept-TRAK's practical utility for understanding how diffusion models compose multiple concepts from training data, providing insights valuable for copyright analysis, model debugging, and interpretability research.

### D.3 Controlled evaluation: CelebA-HQ (Qualitative)

In Figure 13, we present qualitative results for concept-level attribution on the CelebA-HQ dataset. This replicates the trends observed in the synthetic dataset: in ID scenarios, images with the same concept as the generated sample can be found through visual similarity alone, but OOD scenarios require isolating individual concepts from compositionally novel outputs, where visual similarity alone fails.

### D.4 Applications of Concept-Level Attribution

Our concept-level attribution method provides valuable insights across multiple domains, as shown in Figure 14. For **copyright protection**, we trace training samples that influenced IP-protected concepts like Mario and Mickey Mouse, addressing provenance concerns. In the realm of **safety**, our method identifies training samples contributing to sensitive concepts, enabling targeted data curation for responsible AI development. For **model debugging**, Concept-TRAK pinpoints sources of both desirable features and problematic outputs, enhancing our understanding of prompt engineering. Finally, for **concept learning**, our approach reveals how models acquire complex relational concepts like "hug" and "shake hands.". These applications demonstrate how concept-level attribution provides practical tools for addressing key challenges in generative AI development and governance. Note that these experiments use global concept attribution.

## E Implementation Details

### E.1 Computational Resources

All experiments were conducted on NVIDIA H100 GPUs with 80GB memory. To reduce computational costs, all experiments were performed using fp16 precision.

Influence function-based attribution methods consist of two computational stages: (1) a one-time preprocessing cost of computing training gradients for all training samples, and (2) a lightweight per-query step that computes utility gradients for each concept or query. For SD v1.4 (LAION-100K), our concrete measurements are as follows:

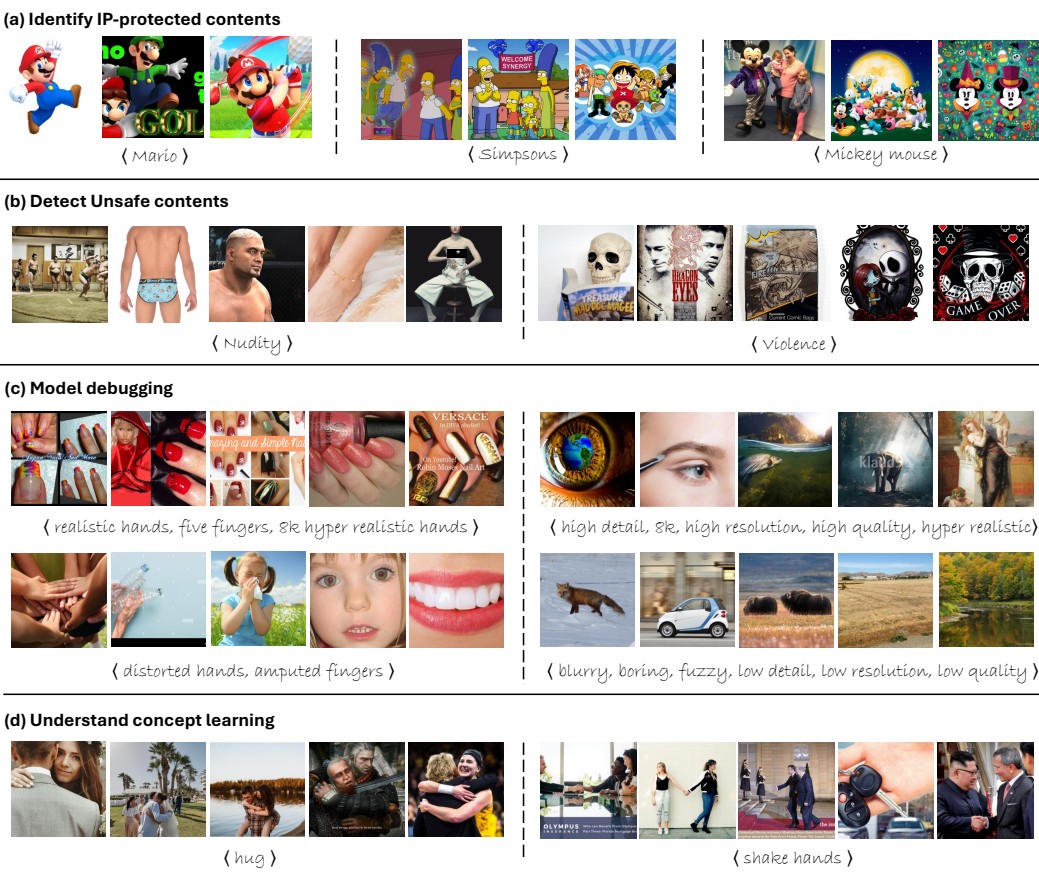

Figure 14: Applications of concept-level attribution across diverse tasks. (a) Identifying training sources of IP-protected characters. (b) Detecting origins of sensitive content for safety governance. (c) Tracing sources of desirable and problematic features for model debugging. (d) Revealing how models acquire relational concept understanding.

**Training gradient computation**   The TRAK baseline requires approximately 16 GPU hours, while Concept-TRAK requires 32 GPU hours, due to additional operations including DDIM inversion, and prompt-level guidance.

**Per-concept attribution**   Once gradients are cached, the TRAK baseline requires 1 minute per query, while Concept-TRAK requires 3 minutes per query.

## E.2   SYNTHETIC DATASET

**Data Generation** We randomly sample shape and color attributes and place them at random positions within the image canvas. Image resolution is 64×64. We generate a total of 10,000 synthetic images with this procedure.

**Model Training** We observe that dropping class conditions for classifier-free guidance severely harms compositional generalization ability, so we avoid this practice. The model is trained using the Muon optimizer. While Adam optimizer produces qualitatively similar results, we choose Muon for its stability and significantly faster convergence. We train separate ResNet-based classifiers for each concept and continue training until out-of-distribution (OOD) sample generation accuracy reaches 99%. Training hyperparameters follow Jordan et al. (2024): Muon learning rate of 1e-3 for 1000 epochs, with identical momentum and other hyperparameters. For non-matrix parameters, we use Adam optimizer with learning rate 1e-4. We used LightningDiT, state-of-the-art a modernized DiT architecture (Yao et al., 2024) for diffusion model.

**Test Sample Generation** We generate images starting from random noise. Images are regenerated until a separate classifier confirms they match the conditioned concept, with a maximum of 3 regeneration attempts per sample.

**Gradient Computation** *Baseline methods*: For each training sample, we sample 10 different $x_t$ and compute gradients using each method's respective loss function. *Ours*: For the training loss computation, we apply DDIM inversion with guidance scale 2, which we find beneficial for performance. Since we do not train null tokens, we apply CFG by sampling random conditions at each step following Sadat et al. (2024). We hypothesize that applying guidance during DDIM inversion removes concept $c$ from $x_0$ and learns tangent vectors that restore this concept, thus positively affecting concept attribution. More detailed analysis and improvements remain interesting future work. For utility loss computation, we do not use CFG.

### E.3 CELEBA-HQ DATASET

**Data Preparation** We use 30,000 images from CelebA-HQ dataset, excluding all samples containing the combination {eyeglasses + male + smiling}. We resize all images to 64×64, to reduce computation.

**Model Training** We follow the identical training recipe as the synthetic dataset. We train separate ResNet-based classifiers for each concept and continue training until OOD sample generation accuracy reaches 95%.

**Test Sample Generation** We generate images starting from random noise. Images are regenerated until a separate classifier confirms they match the conditioned concept, with a maximum of 3 regeneration attempts per sample.

**Gradient Computation** We follow the identical gradient computation recipe as described for the synthetic dataset.

### E.4 ABC BENCHMARK

This subsection presents the detailed experimental setup for our evaluation of the AbC benchmark.

**Benchmark Construction** To address more realistic data attribution scenarios, we modify the original AbC benchmark setup. Rather than fine-tuning model parameters on customization data, we freeze the base model parameters and train only special tokens through textual inversion (Gal et al., 2022). Following Wang et al. (2023b), we create 20 special tokens corresponding to 20 customization concepts. For each special token, we generate 20 images, resulting in 400 total generated images for data attribution evaluation.

We perform textual inversion using the default hyperparameters provided by the diffusers library: AdamW optimizer with learning rate $5.0 \times 10^{-4}$, batch size 4, and training epochs 3000.

**Baseline Methods** Both TRAK, D-TRAK, and DAS need to specify a regularization hyperparameter $\lambda$. To be more specific, in TRAK (Park et al., 2023a), we approximate the inverted projected Hessian as $\mathbf{H}_P^{-1} \approx (\mathbf{F}_P + \lambda I)^{-1}$, where $\mathbf{F}_P = \frac{1}{N} \sum_k G^\top G$ and $G_{ij} = \nabla_{\theta_j} L(x_i; \theta)$. The regularization $\lambda$ is applied to make sure to $\mathbf{H}_P \approx \mathbf{F}_P + \lambda I$ is invertible in practice. On the other hand, this regularization makes TRAK-based data attribution effectively ignore components with small eigenvalues, significantly impacting attribution performance (Choe et al., 2024).

Previous work recommends $\lambda^* = 0.1 \times \text{mean}(\text{eigenvalues}(\mathbf{F}_P))$ (Grosse et al., 2023). For a fair comparison, we perform a hyperparameter sweep for TRAK, D-TRAK, DAS and ours across $\lambda \in [\lambda^* \times 10^{-4}, \lambda^* \times 10^4]$ and report the best performance achieved.

**Concept-TRAK Configuration** Our method focuses on measuring contributions to specific concepts $c$. We use the synthetic image generation prompt as $c^+$ (e.g., "a tattoo of $\langle V \rangle$ snake") and remove only the special token to create $c^-$ (e.g., "a tattoo of snake"). We used $c^+$ as the base prompt $c$. We set the regularization parameter to the recommended value $\lambda^* = 0.1 \times \text{mean}(\text{eigenvalues}(\mathbf{F}_P))$ (Grosse et al., 2023) and use 1024 $x_t$ from random trajectories sampled from DDIM sampler for gradient computation.

### E.5 QUALITATIVE RESULTS

For the qualitative case studies presented in Appendix D.4, we maintain consistent experimental settings with the AbC benchmark evaluation. The regularization parameter is set to $\lambda = 0.1 \times$ mean(eigenvalues($\mathbf{F}_P$)) across all methods.

Below are the prompts used for each concept-level attribution task:

- **IP-protected content**:
  - Mario:
    - $c$ "character, Mario"
    - $c_- = $ "character"
  - Mickey Mouse:
    - $c = $ "character, Mickey Mouse"
    - $c_- = $ "character"
  - Simpsons:
    - $c = $ "character, Simpsons"
    - $c_- = $ "character"
- **Unsafe content**:
  - Nudity:
    - $c = $ "person, nude, naked"
    - $c_- = $ "person"
  - Violence:
    - $c = $ "zombie, bloody, violence, gore, horror, grotesque, skull, fear, death, creature"
    - $c_- = $ ""
- **Model debugging**:
  - Distorted hands:
    - $c = $ "hands, distorted hands, amputated fingers"
    - $c_- = $ "realistic hands, five fingers, 8k hyper realistic hands"
  - High quality images:
    - $c$ = "high detail, 8k, intricate, detailed, high resolution, high res, high quality, hyper realistic"
    - $c_-$ = "blurry, boring, fuzzy, low detail, low resolution, low res, low quality"
- **Concept learning**:
  - Hug:
    - $c = $ "people hug each other"
    - $c_- = $ "people"
  - Shake hands:
    - $c = $ "people shake their hands"
    - $c_- = $ "people"

**Note on Model debugging**  For the model debugging study, we perform bidirectional attribution by swapping $c$ and $c_-$ to identify both positive and negative influences. This allows us to trace training samples that contribute to both problematic and desirable generation.

## F  ALGORITHM

In this section, we provide detailed algorithms for computing training gradients using the train loss Eq. (3) (Algorithm 1) and utility gradients using utility loss Eq. (4) (Algorithm 2) used in *Concept-TRAK*. The key computational steps highlighted in red show the guidance terms that distinguish our approach from standard methods.

**Algorithm 1** Train loss $\mathcal{L}_{\text{train}}$

**Require:** $x_0^i$, $N$, $\{\bar{\alpha}_t\}_{t=0}^T$
1: **for** $n = 1$ **to** $N$ **do**
2:      $x_t^i \leftarrow \text{DDIMinv}(x_0^i, 0 \rightarrow \frac{nT}{N})$, $t \leftarrow \frac{nT}{N}$
3:      $\hat{x}_0^i \leftarrow \frac{1}{\sqrt{\bar{\alpha}_t}} \left( x_t^i - \sqrt{1 - \bar{\alpha}_t}\, \epsilon_\theta(x_t^i) \right)$
4:      $\color{red}{\delta_{\text{DPS}} \leftarrow -\nabla_{x_t} \|\hat{x}_0^i - x_0^i\|_2^2}$
5:      $\tilde{\epsilon}_\theta(x_t^i) \leftarrow \text{sg}[\epsilon_\theta(x_t^i) - \color{red}{\delta_{\text{DPS}}}]$
6:      $\mathcal{L}_{\text{DPS}} \leftarrow \|\tilde{\epsilon}_\theta(x_t^i) - \epsilon_\theta(x_t^i)\|_2^2$
7:      $g_n \leftarrow \nabla_\theta \mathcal{L}_{\text{DPS}}$
8: **end for**
9: $g \leftarrow \frac{1}{N} \sum_{n=1}^N g_n / \|g_n\|_2$
10: **return** $g$

**Algorithm 2** Utility loss $\mathcal{L}_{\text{concept}}$

**Require:** $N$, $\{\bar{\alpha}_t\}_{t=0}^T$, $\{\eta_t\}_{t=0}^T$
1: **for** $n = 1$ **to** $N$ **do**
2:      **if** local attribution **then**
3:          $x_T \leftarrow$ Noise used to generate $x_0^{\text{test}}$
4:      **else**
5:          $x_T \sim \mathcal{N}(0, I)$
6:      **end if**
7:      $t \sim \text{Uniform}(0, T)$
8:      $x_t \leftarrow \text{DDIM}(x_T \rightarrow t)$
9:      $\hat{x}_0 \leftarrow \frac{1}{\sqrt{\bar{\alpha}_t}} \left( x_t - \sqrt{1 - \bar{\alpha}_t}\, \epsilon_\theta(x_t) \right)$
10:     $\color{red}{\delta_{\text{Reward-DPS}} \leftarrow \nabla_{x_t} R(x_t)}$
11:     $\tilde{\epsilon}_\theta(x_t) \leftarrow \text{sg}[\epsilon_\theta(x_t) - \color{red}{\delta_{\text{Reward-DPS}}}]$
12:     $\mathcal{L}_{\text{Reward-DPS}} \leftarrow \|\tilde{\epsilon}_\theta(x_t) - \epsilon_\theta(x_t)\|_2^2$
13:     $g_n \leftarrow \nabla_\theta \mathcal{L}_{\text{Reward-DPS}}$
14: **end for**
15: $g \leftarrow \frac{1}{N} \sum_{n=1}^N g_n / \|g_n\|_2$
16: **return** $g$

