# OpenReview forum: "Concept-TRAK: Understanding how diffusion models learn concepts through concept attribution"
_ICLR.cc/2026/Conference — ICLR 2026 Poster_

### Official Review · Reviewer_Xw2f · 2025-10-22

**Soundness:** 3
**Presentation:** 3
**Contribution:** 3
**Rating:** 6
**Confidence:** 2

**Summary:**

The paper introduces Concept-TRAK, which attributes training samples that taught a specific concept in diffusion models by aligning a per-sample training gradient with a concept gradient. They first introduce losses that operate through reward gradients in the tangent space and construct an influence function framework based on these losses. Evaluations are performed on controlled synthetic datasets as well as CelebA dataset and AbC benchmark for data attribution evaluations.

**Strengths:**

The paper proposes a novel problem where the focus is to pinpoint which training images taught a specific concept rather than just influencing an entire image.

**Weaknesses:**

1. The evaluations are very narrow and limited with respect to the concepts used. In real-world settings, the concepts can be highly compositional and overlapping, whereas the concepts used in the paper are simple and clean, which doesn’t reflect the real-world setting.

2. The paper lacks a human evaluation for the applications shown in Fig. 8. It is important to analyse if the retrieved samples actually reflect the target concept in these cases. Adding a human judgment study would strengthen the claims.

**Questions:**

1. I did not see any experiments that show the computational effectiveness of the approach. How much time does the approach take for a single concept?

---

> ### Author Response · Authors · 2025-11-27
>
> We thank the reviewer for acknowledging that concept-level attribution represents a novel problem formulation.
>
> ---
>
> > [W1] Limited concept diversity: evaluations use simple, non-overlapping concepts that don't reflect real-world complexity
>
> Thank you for this important feedback! The **Global Rebuttal** and **Appendix D.2** presents comprehensive new experiments addressing:
> - **Semantically similar concepts** (cat, tiger, jaguar, leopard, cheetah)
> - **Compositional concepts** at three difficulty levels, from simple object style composition to complex multi-concept composition.
>
> These experiments demonstrate that Concept-TRAK successfully handles real-world complexity where baseline methods fail. We greatly appreciate this suggestion, which substantially strengthened our evaluation.
>
> > [W2] Lack of human evaluation for retrieved samples
>
> Thank you for this excellent suggestion! While not ground-truth evaluation, your recommendation enabled us to conduct extensive quantitative evaluation of real-world scenarios. Due to volume of the experiment, we employed **Qwen3-VL** (a strong open-source VLM) instead of human evaluation. The model assesses whether retrieved images contain the target concept.
>
> Detailed quantitative results are presented in the **Global Rebuttal** and **Appendix D.2**.
>
> > [Q1] Computational effectiveness of the approach
>
> **Computational cost breakdown**: Influence function-based attribution methods consist of two computational stages: (1) a one-time preprocessing cost of computing training gradients for all training samples, and (2) a lightweight per-query step that computes utility gradients for each concept or query. For SD v1.4 (LAION-100K), our concrete measurements are as follows:
>
> - Training gradient computation: TRAK baseline requires approximately 16 GPU hours, while Concept-TRAK requires around 32 GPU hours due to additional DDIM inversion and guidance computation.
> - Per-concept attribution: TRAK baseline requires under 1 minute, while Concept-TRAK requires around 3 minutes
>
> We want to note that our primary contribution is concept-level attribution, a new task formulation and the first effective influence-based solution for this setting, not a proposal for improving the computational efficiency of data attribution itself. The computational profile of Concept-TRAK stems directly from the underlying influence-function framework and is therefore orthogonal to our core contribution. We think efficiency improvements for concept-level attribution remain interesting future work.
>
> We have added detailed analysis of computational cost of Concept-TRAK in Appendix E.1.
>
> ---
>
> We sincerely thank you for the thoughtful suggestion. Thanks to your insights, our work has become significantly stronger, supported by expanded experiments across diverse real-world scenarios and quantitative evaluation using VLM-based assessment. Once again, we deeply appreciate your time and valuable feedback.

---

### Official Review · Reviewer_KaYm · 2025-10-28

**Soundness:** 3
**Presentation:** 3
**Contribution:** 3
**Rating:** 4
**Confidence:** 2

**Summary:**

This paper introduces Concept-TRAK, a novel framework for concept-level attribution in diffusion models. The method reformulates influence functions through a reward optimization framework, where a concept-aware loss is defined using gradients along the tangent space of the diffusion manifold. The authors leverage Explicit Score Matching to approximate concept-relevant gradients and compute per-sample attributions without re-training the diffusion model. Empirical studies on synthetic datasets, CelebA-HQ, and the AbC benchmark demonstrate that Concept-TRAK outperforms baselines (TRAK, D-TRAK, DAS) both in in-distribution and out-of-distribution concept attribution, achieving substantially higher performance.

**Strengths:**

1. The paper moves beyond image-level attribution toward concept-level interpretability, a direction of genuine novelty and relevance for auditing large-scale generative models.
2. The integration of reward optimization and influence functions is conceptually elegant.
3. The method has potential applications in copyright auditing and transparency for generative AI, aligning with broader research interests in responsible model development.

**Weaknesses:**

1. The approach likely has high computational complexity due to Hessian approximations and multiple gradient evaluations per sample.
2. The synthetic dataset uses only a 2×2 factor combination; CelebA-HQ omits a single triple attribute; AbC relies on textual inversion with a frozen base model. These experiments demonstrate the idea but fall short of realistic large-scale diffusion scenarios involving diverse prompts and complex concept interactions.
3. The method is described as “training a diffusion model on a reward-shaped distribution,” but in reality, Concept-TRAK operates post-hoc on pretrained models without updating parameters. This wording could mislead readers regarding optimization vs. analysis.

**Questions:**

1. What is the time and memory cost of running Concept-TRAK on a large model like Stable Diffusion XL or SD 2.1?
2. How does attribution behave when concepts overlap semantically?
3. In prompts with multiple intertwined concepts (e.g., “a cat in Van Gogh style”), how is the negative concept constructed, and how does Concept-TRAK avoid misattributing correlated concepts (cat vs. tiger, style vs. object)?

---

> ### Author Response · Authors · 2025-11-27
>
> We sincerely thank the reviewer for acknowledging that our research direction is genuinely novel and that our method is conceptually elegant.
>
> ---
>
> > [W1] Computational complexity: Hessian approximations and multiple gradient evaluations
>
> Our method is built on TRAK, which approximates the Hessian using the Fisher Information Matrix (FIM). Given pre-computed training gradients, the FIM can be computed with negligible computational overhead through a single matrix multiplication. While this detail was originally documented in Appendix E.4 (L1039), we have now added this clarification to the main manuscript at Section 3.4 (L319) to improve clarity and reduce confusion about computational requirements.
>
> > [W2] diverse prompts and complex concept interactions
> > [Q2] How does attribution behave when concepts overlap semantically?
> > [Q3] Handling multiple intertwined concepts and avoiding misattribution
>
> Thank you for these excellent suggestions! The **Global Rebuttal** and **Appendix D.2** presents extensive new experiments demonstrating that Concept-TRAK substantially outperforms baseline methods when handling:
> - **Similar concepts** (e.g., cat vs. tiger vs. jaguar)
> - **Complex compositional concepts** at multiple difficulty levels
>
> We particularly recommend reviewing the qualitative results for complex prompts such as "An astronaut riding a horse on Mars". Thank you for helping us significantly strengthen our experiments!
>
> > [W3] Misleading wording regarding "optimization vs. analysis"
>
> Thank you for this important clarification. We have revised the manuscript to emphasize that our method define the loss function to analyze gradient directions rather than performing optimization. The updated text clarifies this distinction (L229):
>
> > To analyze the gradient direction toward this reward-shaped distribution, we define a loss based on Explicit Score Matching (ESM) (...)
>
> > [Q1] Time and memory cost for large models (SDXL, SD 2.1)
>
>
> **Computational cost breakdown**: Influence function-based attribution methods consist of two computational stages: (1) a one-time preprocessing cost of computing training gradients for all training samples, and (2) a lightweight per-query step that computes utility gradients for each concept or query. For SD v1.4 (LAION-100K), our concrete measurements are as follows:
>
> - Training gradient computation: TRAK baseline requires approximately 16 GPU hours, while Concept-TRAK requires around 32 GPU hours due to additional DDIM inversion and guidance computation.
> - Per-concept attribution: TRAK baseline requires under 1 minute, while Concept-TRAK requires around 3 minutes
>
> Although we currently lack the resources to directly measure SD v2.1 or SDXL, considering the model size, SD v2.1 remain within the same order of magnitude as SD v1.4. For SDXL, we conservatively estimate roughly 4× higher gradient computation time, consistent with its 4× larger parameter count.
>
> **Memory cost breakdown**: the peak GPU memory required to compute gradients per sample for SD1.4v is 18.25 GB. When scaling to SDXL, we conservatively estimate an approximate 4× increase in peak memory usage (∼73 GB), reflecting its larger parameter count. Importantly, note that we have not yet applied engineering techniques for peak-memory reduction, such as gradient checkpointing.
>
> We have added detailed analysis of computational cost of Concept-TRAK in Appendix E.1.
>
> ---
>
> We sincerely appreciate your thoughtful suggestions and insightful questions. Your feedback has helped us substantially strengthen both our experiments (especially real-world scenarios) and the clarity of our writing. Thank you for taking the time to provide such valuable guidance.

---

### Official Review · Reviewer_vod1 · 2025-11-01

**Soundness:** 2
**Presentation:** 2
**Contribution:** 3
**Rating:** 6
**Confidence:** 3

**Summary:**

This paper introduces a novel data attribution method for diffusion models designed to identify which training samples influenced a specific concept within a generated image. The key innovation is the use of specialized, reward-based loss functions for both the training sample and the target concept, which are designed to capture concept-relevant directions in the model's tangent space. Experiments show that Concept-TRAK substantially outperforms prior methods in these concept-level attribution scenarios.

**Strengths:**

- The paper addresses a significant and overlooked limitation in existing data attribution for diffusion models, which is attributing influence for a *specific* concept.
- The paper conducts controlled experiments and shows that Concept-TRAK performs well in both ID and OOD settings.

**Weaknesses:**

- If I understand correctly, other methods only allow attribution to a specific generated image. How do you evaluate other methods in the experiments? A naive way is to find influential training data to a set of generated images containing the target concept. Is this how you conduct the experiment? If not, the paper should include that as well.
- The paper does not specify which model version is evaluated in the paper. A thorough experiment with many model types, such as SD 1.4, SD v2, SD XL, Flux, DeepFloyd, etc. would strengthen the claim

**Questions:**

Please see Weaknesses.

---

> ### Author Response · Authors · 2025-11-27
>
> We sincerely thank the reviewer for acknowledging that our work addresses a **significant and overlooked limitation** in existing data attribution for diffusion models.
>
> ---
>
> > [W1] Additional baseline: **set** of generated images containing the target concept
>
> Thank you for this excellent suggestion! We have implemented this baseline and present comprehensive results in the **Global Rebuttal**. Our experiments demonstrate that Concept-TRAK outperforms the proposed D-TRAK/DAS-based global concept attribution across all benchmarks (AbC / Toy / CelebA-HQ) and achieves higher accuracy in real-world scenarios. We have added detailed discussion in **Appendix D.1**.
>
> > [W2] Model version specification and evaluation across multiple T2I architectures
>
> In the main manuscript (L423), we specified that we used **Stable Diffusion v1.4**. While we agree that evaluation across multiple T2I models (SD 2.0, SDXL, Flux, DeepFloyd, etc.) would strengthen our claims, there are significant practical constraints:
>
> **Computational challenges:**
> - Data attribution requires access to the model's training data
> - Influence function-based methods require computing training gradients for all training samples
> - This imposes substantial computational requirements, particularly for larger models
>
> **Method generality:**
> We emphasize that our method is built on influence functions, which are model-agnostic by design. There are no architectural constraints preventing application to other diffusion models. Validation of our method on additional architectures represents interesting future work.
>
> **Expanded evaluation:**
> To address concerns about generalizability, we have significantly expanded our real-world scenario experiments (see **Global Rebuttal** and **Appendix D.2**), demonstrating Concept-TRAK's capabilities across diverse and challenging cases. These experiments substantially enrich our demonstration of the method's effectiveness.
>
> ---
>
> Once again, we sincerely thank you for the thoughtful review. Thanks to your insights, our work has become significantly stronger with additional suggested baseline methods. We sincerely appreciate the effort and care you put into reviewing our work.

---

### Author Response · Authors · 2025-11-27
**General response**

We sincerely thank all reviewers for their thoughtful and constructive feedback. We are especially grateful that reviewers found our work addresses a significant and overlooked limitation of previous data attribution methods (vod1), and that it is well-motivated and novel (KaYm, Xw2f). Below, we respond to your comments with additional experiments conducted during the rebuttal period.

---

## Additional Experiments Summary

Following the rebuttal, we have added the following figures and sections:

**Additional Baseline (Appendix D.1)**
- Global concept attribution - Toy: Table 4
- Global concept attribution - CelebA-HQ: Table 5
- Global concept attribution - AbC: Table 6

In short, for global concept attribution, concept-trak shows comparable performance to the baseline methods that reviewer vod1 proposed.

**Real-world scenario (Appendix D.2)**
- Similar concept - Figure 7, Table 7, 8
- Compositional concept - Figure 8, 9,Table 9, 10
- Complex prompt - Figure 10

In short, across a variety of real-world scenarios, concept-trak demonstrates superior concept attribution performance compared to baseline methods.

These sections and figures have been visually highlighted in blue so reviewers can quickly identify the updates.

---

## 1. Additional Baseline: Global Concept Attribution

> *"A naive way is to find influential training data for a **set** of generated images containing the target concept."* (vod1)

Thank you for this excellent suggestion! The proposed approach uses the utility gradient averaged across multiple images containing concept $c$:

$$L_{\text{concept}} = \mathbb{E}_{x_0 \sim p(x|c)}[L(x_0; c)]$$

Assuming $L$ approximates $p(x|c)$, this closely aligns with our concept attribution utility definition in Section 3.1:

$$p_\theta(c) = \mathbb{E}_{x_0 \sim p(x|c)}[p(c|x)]$$

Since $x_0 \sim p(x|c^{\text{target}})$ is sampled from the model's generative distribution rather than being a single image, this approach only enables **global concept attribution**. In contrast, Concept-TRAK additionally enables **local concept attribution** for specific generated images.

---

### Experimental Setup

We conducted experiments on all benchmarks (AbC / Toy / CelebA-HQ):

- **Toy / CelebA-HQ**: Fixed target concept $c$, randomized other attributes, generated 256 images from $p(x|c)$ to compute utility gradient
- **AbC**: Generated 256 images using prompts provided by AbC benchmark to compute utility gradient

---

### Results

**Toy: Global baseline**

|       | Concept-TRAK | DAS | DTRAK |
|-------|--------------|-----|-------|
| Shape | 1.0          | 1.0 | 1.0   |
| Color | 0.9          | 0.7 | 1.0   |
| Avg.      | 0.95          | 0.85 | **1.0**   |

**CelebA-HQ: Global baseline**

|            | Concept-TRAK | DAS | DTRAK |
|------------|--------------|-----|-------|
| Eyeglasses | 1.0          | 1.0 | 1.0   |
| Male       | 1.0          | 1.0 | 1.0   |
| Smile      | 1.0          | 1.0 | 1.0   |
| Avg.      | **1.0**          | **1.0** | **1.0**   |

**AbC: Global baseline**

|        | Concept-TRAK | DAS  | DTRAK |
|--------|--------------|------|-------|
| Object | 0.89         | 0.98 | 0.95  |
| Style  | 0.93     | 0.81 | 0.83  |
| Avg.  | **0.91**     | 0.895 | 0.89  |

As shown in the tables, Concept-TRAK's global concept attribution achieves comparable performance on Toy and CelebA-HQ, and better average performance on AbC, relative to prior methods. We added this experiment results to the Appendix D.1. Thank you again for suggesting this experiment!

---

> ### Author Response · Authors · 2025-11-27
>
> ## 2. Real-world Scenario: Similar / Complex Concepts
>
> ---
>
> ### Evaluation Methodology
>
> > (Xw2f) *"The paper lacks a human evaluation for the applications (...). Adding a human judgment study would strengthen the claims."*
>
> Thank you for suggesting an evaluation approach for real-world applications! While not as rigorous as our controlled benchmarks (AbC / Toy / CelebA-HQ), verifying whether retrieved images contain the target concept provides a valuable sanity-check level evaluation.
>
> Given the volume of experiments below, human judgment was not feasible. Instead, we used **Qwen3-VL-8B** (a strong open-source VLM) for evaluation. Specifically, we queried whether retrieved images contain the concept; if not, we counted it as inaccurate concept attribution. Similar to the AbC benchmark, we report **Precision@10**.
>
> For certain images, the number of training images contributing to a specific concept (e.g., "cat") may be fewer than 10. Therefore, the upper bound for local concept attribution performance is not necessarily precision = 1.0. We recommend interpreting these metrics as **relative performance indicators** rather than absolute scores.
>
> ---
>
> ## Similar Concepts
>
> > *"How does Concept-TRAK avoid misattributing correlated concepts (cat vs. tiger)?"* (KaYm)
> >
> > *"How does attribution behave when concepts overlap semantically?"* (KaYm)
> >
> > *"In real-world settings, the concepts can be highly overlapping."* (Xw2f)
>
> We first address whether the method can correctly distinguish between semantically similar concepts that share visual features. We select five big cat species: *cat, tiger, jaguar, leopard, and cheetah*, that are visually similar.
>
> For local attribution, we generate 8 images for each concept using simple prompts (e.g., "a cat") with different random seeds (0 to 7). For each generated image, we perform concept attribution and measure Precision@10, checking whether the top-10 retrieved training samples contain the target concept (e.g., cat). For global attribution, we generate 256 images per concept and compute set-level attribution, then evaluate whether the retrieved training samples contain the correct concept.
>
> ---
>
> Results are as follows:
>
> **Local Attribution (avg across 8 seeds)**
>
> |         | Concept-TRAK | DAS   | DTRAK | TRAK  |
> |---------|--------------|-------|-------|-------|
> | cat     | **0.925**    | 0.000 | 0.000 | 0.000 |
> | tiger   | **0.662**    | 0.000 | 0.000 | 0.000 |
> | jaguar  | **0.188**    | 0.000 | 0.000 | 0.000 |
> | leopard | **0.300**    | 0.000 | 0.000 | 0.087 |
> | cheetah | **0.325**    | 0.000 | 0.000 | 0.000 |
> | Average | **0.480**    | 0.000 | 0.000 | 0.017 |
>
> **Global Attribution**
>
> |         | Concept-TRAK | DAS       | DTRAK     | TRAK  |
> |---------|--------------|-----------|-----------|-------|
> | cat     | **1.000**    | **1.000** | **1.000** | 0.900 |
> | tiger   | **1.000**    | 0.800     | 0.800     | 0.800 |
> | jaguar  | **0.400**    | 0.300     | 0.200     | 0.200 |
> | leopard | **0.600**    | **0.600** | **0.600** | 0.300 |
> | cheetah | **0.500**    | 0.300     | 0.300     | 0.300 |
> | Average | **0.700**    | 0.600     | 0.580     | 0.500 |
>
> For single concept attribution, Concept-TRAK significantly outperforms all baseline methods. Qualitative results in Figure 7 reveal that for relatively rare concepts like "jaguar," the model sometimes retrieves images with similar visual features (e.g., tiger, leopard) as contributors, suggesting these similar examples did influence the learned concept representation.

---

> ### Author Response · Authors · 2025-11-27
>
> ## Complex Concepts
>
> > (KaYm) *"How does Concept-TRAK avoid misattributing correlated concepts (style vs. object)?"*
> >
> > (Xw2f) *"In real-world settings, the concepts can be highly compositional."*
>
> Second, we address whether the method can accurately capture training samples contributing to each concept when generated images contain complex and diverse concepts. We prepared three experiments:
>
> ---
>
> ### Composition Level 1: Common Object + Style
>
> We generate images combining common objects with artistic styles using prompts of the form "{object} in the style of {style}". Specifically, we use two objects (cat, dog) and two artistic styles (graffiti art, stained glass), generating 8 images per object-style combination for a total of 32 images. For each image, we perform separate concept attribution for the object concept and the style concept. For global attribution baselines, we generate 256 images from each full prompt and perform set-level attribution, then separately evaluate whether retrieved samples contain the object or style.
>
> ---
>
> **Results**
>
> |                 | Concept-TRAK | DAS   | DTRAK | TRAK  |
> |-----------------|--------------|-------|-------|-------|
> | Local (Object)  | **0.934**    | 0.025 | 0.025 | 0.034 |
> | Local (Style)   | **0.919**    | 0.047 | 0.047 | 0.013 |
> | Global (Object) | **1.000**    | 0.000 | 0.000 | 0.125 |
> | Global (Style)  | **0.950**    | 0.925 | 0.925 | 0.850 |
>
> Concept-TRAK substantially outperforms baselines. For **local attribution**, baseline methods completely fail, while Concept-TRAK achieves over 90% accuracy. For **global attribution**, Concept-TRAK also performs better overall. Interestingly, baseline methods appear biased toward style-based attribution, performing relatively well on global style attribution but completely failing on object attribution. For qualitative results, see Figure 8.
>
> ---
>
> ### Composition Level 2: Unique Object + Style
>
> To increase difficulty, we use unique objects (Pikachu, Simpson) paired with famous artist styles (Vincent van Gogh, Pablo Picasso), generating 8 images per combination with different random seeds.
>
>
> **Results**
>
> |                 | Concept-TRAK | DAS   | DTRAK | TRAK  |
> |-----------------|--------------|-------|-------|-------|
> | Local (Object)  | **0.581**    | 0.000 | 0.000 | 0.000 |
> | Local (Style)   | **0.581**    | 0.000 | 0.000 | 0.000 |
> | Global (Object) | **0.725**    | 0.100 | 0.125 | 0.450 |
> | Global (Style)  | **0.775**    | 0.475 | 0.475 | 0.075 |
>
> The increased difficulty reduces Concept-TRAK's performance to 58% for local attribution, yet baseline methods still completely fail (0%), while Concept-TRAK maintains substantial advantages in global attribution as well. For qualitative results, see Figure 9.
>
> ---
>
> ### Composition Level 3: Complex Prompts (Qualitative)
>
> Finally, we showcase interesting cases by performing concept attribution on frequently used prompts from T2I model usecase:
>
> - **"An astronaut riding a horse on Mars"**
>   - Concept list: {An astronaut, a horse, Mars}
> - **"A teddy bear on a skateboard in Times Square"**
>   - Concept list: {A teddy bear, a skateboard, Times Square}
> - **"Avocado chair"**
>   - Concept list: {Avocado, chair}
>
> Qualitative results can be found in **Figure 10**. Interestingly, for "avocado chair," we observe that the method retrieves training images of round-shaped chairs that visually match the generated image, demonstrating fine-grained concept attribution beyond simple object presence.
>
> ---
>
> If reviewers have any interesting prompts you'd like us to evaluate, please let us know! We will add additional qualitative results until Dec 2.

---

### Meta-Review · Area_Chair_1GKs · 2025-12-11

**Summary:**

This paper introduces Concept-TRAK, an influence-function method for concept-level attribution in diffusion image models. Concept-TRAK defines concept-aware losses and computes per-training-sample attributions by aligning training gradients with concept utility gradients in the model’s tangent space. The method leverages explicit score matching approximations to produce efficient per-query attribution without re-training the generative model. Experiments on synthetic / Toy tasks, CelebA-HQ, and the AbC benchmark show improvements.

Most reviewers agree that the problem formulation, i.e., moving from image-level to concept-level attribution, is novel and well-motivated for copyright auditing and transparency. The approach is conceptually elegant, and the reported empirical gains are promising. Several concerns have been raised by the reviewers, such as limited concept diversity, high computational cost, unclear evaluation setting, lack of human evaluation, and scalability.

During the rebuttal phase, the authors added detailed clarification and new results. Specifically, the global baselines suggested by reviewers are provided, and the results show that Concept-TRAK is comparable. Some experiments on semantically similar concepts and compositional tests were added. The implementation details are clarified, and a computational cost breakdown is provided. Remaining concerns are primarily about scalability and generality. Specifically, the method has not been evaluated experimentally on multiple large T2I architectures (only SD v1.4 was measured), the computational costs are non-trivial and only partly estimated for larger models, and the VLM-based evaluation is a proxy rather than human judgments for real-world assessment.

Overall, concept-level attribution could be an important problem, and Concept-TRAK provides a clear, conceptually sound, and empirically effective solution. The authors addressed the major reviewer concerns during rebuttal. To strengthen the paper further before publication, the authors are encouraged to address the remaining concerns.

**Reviewer Concerns:**

Addressed Concerns:

1. How to evaluate other methods for concept attribution (vod1)
2. Clarity about which model was evaluated (vod1)
3. Handling semantically similar concepts and compositional prompts (KaYm, Xw2f)
4. Computational cost breakdown (KaYm, Xw2f)

Outstanding Concerns:

1. Empirical evaluation on multiple / larger T2I architectures (vod1, KaYm)
2. Human evaluation for real-world scenarios (Xw2f)

**Reviewer Scores:**

Although the authors provided clarification and new results, part of key concerns remains.

* Reviewer vod1 is likely to maintain 6.
* Reviewer KaYm is likely to maintain 4.
* Reviewer Xw2f is likely to maintain 4.

---

### Decision · Program_Chairs · 2026-01-26

Accept (Poster)